# Ingestible pills reveal gastric correlates of emotions

**Giuseppina Porciello[1,2]\*, Alessandro Monti[1], Maria Serena Panasiti[1,2], Salvatore Maria Aglioti[2,3]\***

[1]Department of Psychology, Sapienza Università di Roma, Rome, Italy; [2]IRCCS Fondazione Santa Lucia Research Hospital, Rome, Italy; [3]Sapienza Università di Roma and Center for Life Nano- & Neuro-Science, Fondazione Istituto Italiano di Tecnologia, Rome, Italy

**Abstract** Although it is generally held that gastrointestinal (GI) signals are related to emotions, direct evidence for such a link is currently lacking. One of the reasons why the internal milieu of the GI system is poorly investigated is because visceral organs are difficult to access and monitor. To directly measure the influence of endoluminal markers of GI activity on the emotional experience, we asked a group of healthy male participants to ingest a pill that measured pH, pressure, and temperature of their GI tract while they watched video clips that consistently induced disgust, fear, happiness, sadness, or a control neutral state. In addition to the objective physiological markers of GI activity, subjective ratings of perceived emotions and visceral (i.e. gastric, respiratory and cardiac) sensations were recorded, as well as changes in heart rate (HR), heart rate variability (HRV) and spontaneous eyes blinks as non-gastric behavioral and autonomic markers of the emotional experience. We found that when participants observed fearful and disgusting video clips, they reported to perceive not only cardiac and respiratory sensations but also gastric sensations, such as nausea. Moreover, we found that there was a clear relation between the physiology of the stomach and the perceived emotions. Specifically, when disgusting video clips were displayed, the more acidic the pH, the more participants reported feelings of disgust and fear; the less acidic the pH, the more they reported happiness. Complementing the results found in the deep gastric realm, we found that disgusting stimuli induced a significant increase in HRV compared to the neutral scenarios, and together with fearful video clips a decrease in HR. Our findings suggest that gastric signals contribute to unique emotional states and that ingestible pills may open new avenues for exploring the deep-body physiology of emotions.

**\*For correspondence:**
giuseppina.porciello@uniroma1.it (GP);
salvatoremaria.aglioti@uniroma1.it (SMA)

**Competing interest:** The authors declare that no competing interests exist.

## Editor's evaluation

This important study used a novel method to relate gastric acidity to subjective ratings of emotions induced by video clips. The findings are convincing, have broad implications for the field of emotion research, and open new avenues of research for understanding psychosomatic disorders.

## Introduction

### The neglected role of gastrointestinal (GI) system in emotions

Whether specific patterns of physiological signals coming from visceral organs trigger a unique emotional state is a hotly debated question (*James, 1894*). Somatic theories of emotions posit that afferent physiological signals are essential for experiencing distinct emotional states (*Damasio, 1999*; *Harrison et al., 2010*; *James, 1894*). However, so far the large majority of the studies on the influence

of visceral signals on emotions focused on the role of cardiac and respiratory activity (*Kreibig, 2010*; *Rainville et al., 2006*), neglecting the potential role of the gastrointestinal (GI) tract.

Yet, the GI system is directly connected with the central nervous system via vagal and spinal afferents (*Azzalini et al., 2019*). In turn, different cortical areas directly influence parasympathetic and sympathetic control of the stomach (*Levinthal and Strick, 2020*). Moreover, it has been shown that the slow electrical rhythm (~0.05 Hz) generated in the stomach by the interstitial cells of Cajal interacts with resting-state neural networks (*Rebollo et al., 2018*). This interaction, described as the '*gastric network*' and mostly composed by sensory-motor regions (*Rebollo and Tallon-Baudry, 2022*), seems to be stronger during stress (*Jeanne et al., 2023*), and to go beyond food intake and weight regulatory functions (*Levakov et al., 2021*), possibly playing a role in higher order cognitive and emotional functions (*Azzalini et al., 2019*; *Porciello et al., 2018*; *Rebollo et al., 2021*). That the GI system plays a fundamental role in the experience of emotions has been suggested by studies in which healthy participants were asked to associate basic and complex emotions with the subjective perception of changes taking place in specific parts of the body, including the heart, the lungs, and the GI tract (*Nummenmaa et al., 2018*; *Nummenmaa et al., 2014*). Tellingly, people tend to link changes of activity in the stomach and the bowel with the experience of disgust and happiness (*Nummenmaa et al., 2014*). It is worth noting that, although insightful, these findings derive from mere self-report data. Studies of objective indices of association between deep body activation and emotions are meager. To the best of our knowledge, these few studies show a direct contribution of the GI system to the emotional experience of disgust (*Harrison et al., 2010*; *Shenhav and Mendes, 2014*) using electrogastrography (EGG), a technique that allows recording the myoelectric activity of the stomach through electrodes applied to the skin overlying the epigastric region. Overall, these experiments suggest that the gastric myoelectric dysrhythmias contribute to the emotional experience of disgust (*Harrison et al., 2010*; *Shenhav and Mendes, 2014*) and that disgusting stimuli specifically reduce slow gastric activity (bradygastria) and increase normal (normogastria) and high (tachygastria) gastric activity (*Gianaros et al., 2001*; *Peyrot des Gachons et al., 2011*; *Shenhav and Mendes, 2014*; *Stern et al., 2001*; *Vianna and Tranel, 2006*). In line with these findings, a recent randomized, placebo-controlled study (*Nord et al., 2021*) run on healthy participants showed that oculomotor avoidance (a marker of disgust) evoked by disgusting stimuli significantly decreases after taking domperidone, an antiemetic and prokinetic agent that seems to reduce gastric dysrhythmias. Unfortunately, though, no EGG or any other objective measure of gastric physiology was provided.

## Ingestibles as an innovative method for exploring the deep physiology of emotions

One of the reasons why the endoluminal milieu of the GI system is poorly known is because internal organs are difficult to access and monitor using non-invasive or minimally invasive tools. To overcome this limitation, recent technologies, such as ingestible sensors, have been developed in animal (e.g. *Mimee et al., 2018*) and clinical research for non-invasive diagnostics of diseases affecting the GI system (*Mandsberg et al., 2020*; *Min et al., 2020*; *Rao et al., 2011*), for local drug delivery (*Mandsberg et al., 2020*), and for mapping heart rate, respiratory rate and apneic events, from within the human gastrointestinal tract (*Traverso et al., 2023*). Interestingly, recent findings show that ingestible pills can be used to explore higher-order cognitive and emotional processes (*Mayeli et al., 2023*; *Mayeli et al., 2021*; *Monti et al., 2022*). In particular, we used an ingestible capsule (SmartPill Motility Testing System, Medtronic plc) able to record pH, pressure and inner temperature throughout the entire GI tract (*Saad, 2016*) when participants experienced the illusion of embodying a virtual breathing agent (referred to as the Embreathment, as described in *Monti et al., 2020*; *Cantoni et al., 2024*). We have provided the first evidence (*Monti et al., 2022*) that specific patterns of GI signals co-vary with distinct facets of bodily self-consciousness (i.e. feelings of body location, agency, and disembodiment). Instead, the group led by Khalsa used a vibrating ingestible capsule (Vibrant Ltd) in combination with electroencephalography (EEG), and found an increase in late positive deflections in the event-related potentials (ERPs), maximally located over parieto-occipital electrodes during vibratory gut stimulation. This increase in amplitude correlated with participants' perceptual accuracy in detecting the vibrating stimulation (*Mayeli et al., 2023*; *Mayeli et al., 2021*). In the present study, we used the SmartPills (*Saad, 2016*) while participants watched a validated set of short video clips that reliably induced four basic emotions: fear, disgust, sadness, and happiness (*Tettamanti et al.,*

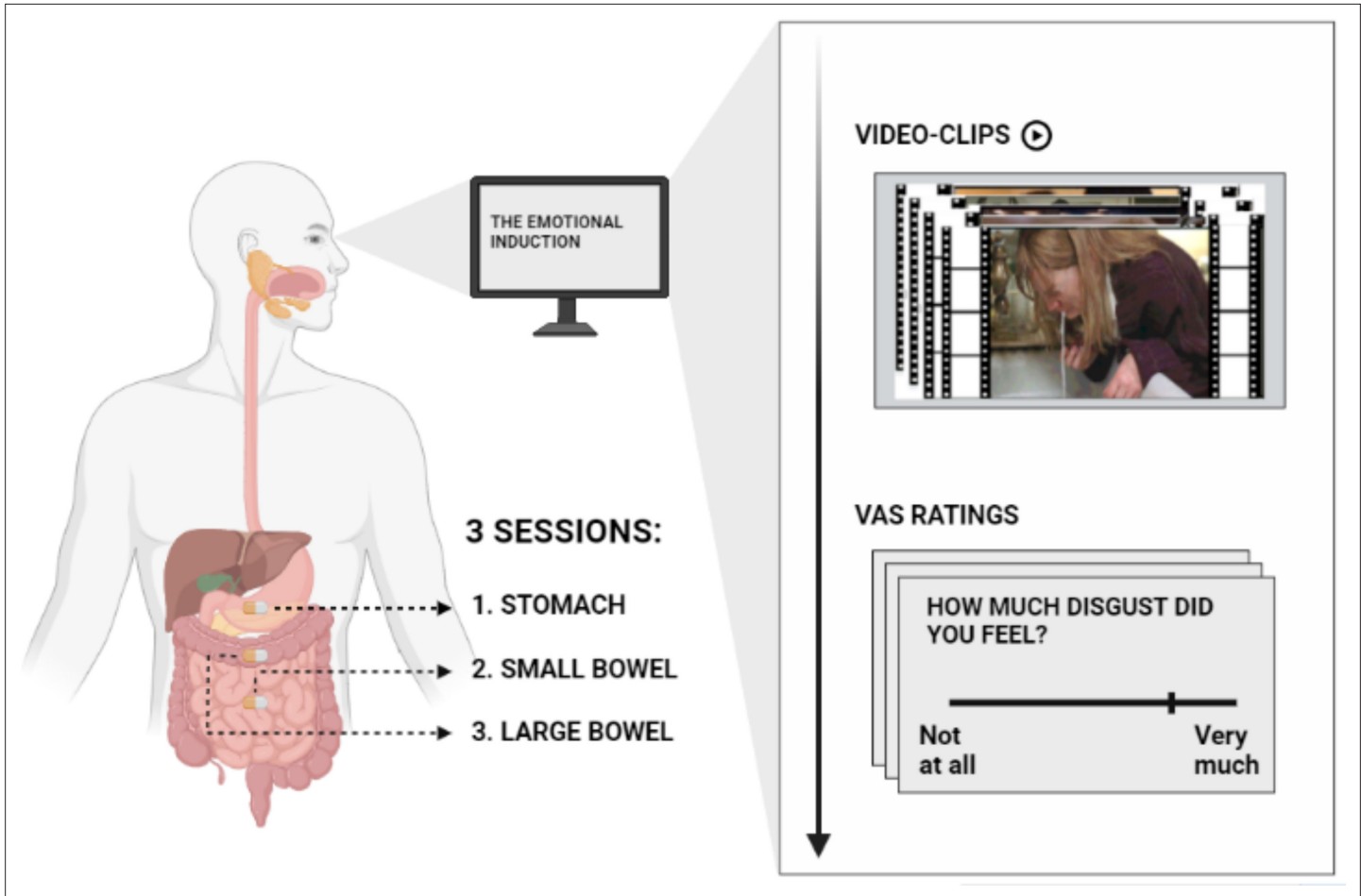

**Figure 1.** Illustration of the emotional induction procedure. Participants were asked to observe five blocks of 24 video clips each. Four blocks were associated with a specific emotion, namely, happiness, disgust, sadness, and fear. An additional block consisting of neutral video clips served as control. Video clips were edited color soundless film excerpts, each lasting ~9 s. They were selected and validated by *Tettamanti et al., 2012*. After each block, participants were asked to respond to an 8-items questionnaire by positioning a computer mouse on 0 ('Not at all') – 100 ('Very much') visuo-analogue scales (VAS). The emotional induction procedure was repeated three times, each corresponding to the ingestible pill position in the stomach (1) in the small (2) and in the large bowel (3).

*2012*), see *Figure 1* for a graphical illustration of the emotional induction procedure. The gastric internal parameters detected through SmartPills were complemented with standard surface EGG (*Yin and Chen, 2013*) that primarily mapped myoelectric activity of the stomach but also cardiac activity, and with vertical electrooculogram that mapped spontaneous eye blinks. Our approach allowed us to identify internally and externally recorded physiological and behavioral markers associated with specific emotional experiences. We also collected self-report ratings of emotions, visceral feelings, and perceived arousal. In this way, we checked the efficacy of our stimuli and measured how the emotional experience induced by the video clips made our participants aware of their visceral state (*Craig, 2003*).

## Hypotheses underlying this study

As far as self-report ratings are concerned, we expected that: (i) in accordance with previous findings (*Tettamanti et al., 2012*) all the emotional video clips would trigger the intended emotion; (ii) basic emotions would be associated with specific visceral sensations mirroring results coming from the cardiac and respiratory domains (e.g. *Rainville et al., 2006*); (iii) participants would report higher gastric sensations after observing disgusting and happy video clips as suggested by *Nummenmaa*

*et al., 2014*; and (iv) there would be no differences in self-reported arousal ratings between the emotional video clips, in line with previous literature (e.g. *Posner et al., 2005*; *Russell and Barrett, 1999*). With respect to the GI physiological signals, our hypotheses relied on a limited number of prior studies that did not make use of the pill technology. Hence, the present study should be considered as the first attempt to explore the neglected contribution of GI internal markers to the emotional experience. On the basis on the published literature on core disgust (*Harrison et al., 2010*; *Nord et al., 2021*; *Shenhav and Mendes, 2014*), we hypothesized that the subjective experience of disgust during the observation of the disgusting video clips would be linked to individual gastric rhythm as indexed by the EGG traces and the GI pressure recorded by the ingestible pill. Moreover, based on single case studies (*Bennett and Venables, 1920*; *Wolf and Wolff, 1947*), we also predicted that the subjective experience of disgust, fear, and sadness would be associated with more acidic pH in the stomach. Based on the literature showing that externally recorded oral or axillary body temperature increases after the observation of disgusting and stressful stimuli (*Marazziti et al., 1992*; *Stevenson et al., 2012*), we predicted that, compared with happy and neutral emotional states, disgust, fear, and sadness, triggered by the corresponding video clips, would be associated with an increase of GI temperature.

Furthermore, building upon literature that proposes specificity in autonomic nervous system responses associated with distinct subtypes of emotions (as extensively reviewed by *Kreibig, 2010*) and supported by experimental findings revealing that different rates of spontaneous eye blinks rates correspond to distinct emotional categories (e.g. *Maffei and Angrilli, 2019*), we also expected variations in heart rate (HR), heart rate variability (HRV), and the amount of spontaneous eye blinks as additional non-gastric indicators of distinct emotional experiences.

## Results
### Subjective emotional experience
#### Capsule in the stomach (Session 1)

To verify the effectiveness of our video clips in eliciting distinct emotional states, we conducted a Friedman ANOVA to compare the perceived emotional experiences (i.e. disgust, fear, happiness, and sadness) triggered by the five types of video clips (varying for their content, i.e. disgusting, fearful, happy, sad, and neutral) during the first session (namely when the capsule was in the stomach). In agreement with the hypotheses, we found that Friedman ANOVA was statistically significant ($\chi^2$ (19)=296.91; p<0.0001), suggesting that participants perceived different emotions after observing the different content of the video clips. Specifically, planned post-hoc Bonferroni-corrected Wilcoxon matched-pairs tests showed that disgust was primarily perceived after watching disgusting video clips, that is the VAS ratings of perceived disgust given after observing disgusting video clips were higher than those given after observation of all the other video clips (all Zs ≥4.372; all ps ≤0.0001). Fear was primarily perceived after watching fearful video clips, that is the VAS ratings of perceived fear given after fearful video clips were higher than after the other video clips (all Zs ≥2.972; all ps ≤0.003). Happiness was primarily perceived after watching happy video clips, that is the VAS ratings of perceived happiness given after happy video clips were higher than after the other video clips (all Zs ≥4.445; all ps ≤0.0001). Finally, sadness was primarily perceived after watching sad video clips, that is the VAS ratings of perceived sadness given after sad video clips were higher than the other video clips (all Zs ≥4.422; all ps ≤0.0001). All the significant post-hoc comparisons are shown in *Figure 2*.

Due to the fact that results on subjective emotional experience collected in session 2 and session 3 mirrored those of session 1, they are described and plotted in the *Appendix 1—figures 1 and 2*.

### Subjective visceral experience
#### Capsule in the stomach (Session 1)

To test the hypothesis that the perceived visceral experience (i.e. gastric, cardiac, respiratory sensations and arousal) varied according to the emotional state triggered by the five types of video clip, and specifically that gastric sensations characterize more disgust and happiness, as suggested by literature (*Nummenmaa et al., 2014*), we performed a Friedman ANOVA. The Friedman ANOVA was statistically significant ($\chi^2$(19)=323.399; p<0.0001), suggesting that participants perceived different visceral sensations after observing the different content of the video

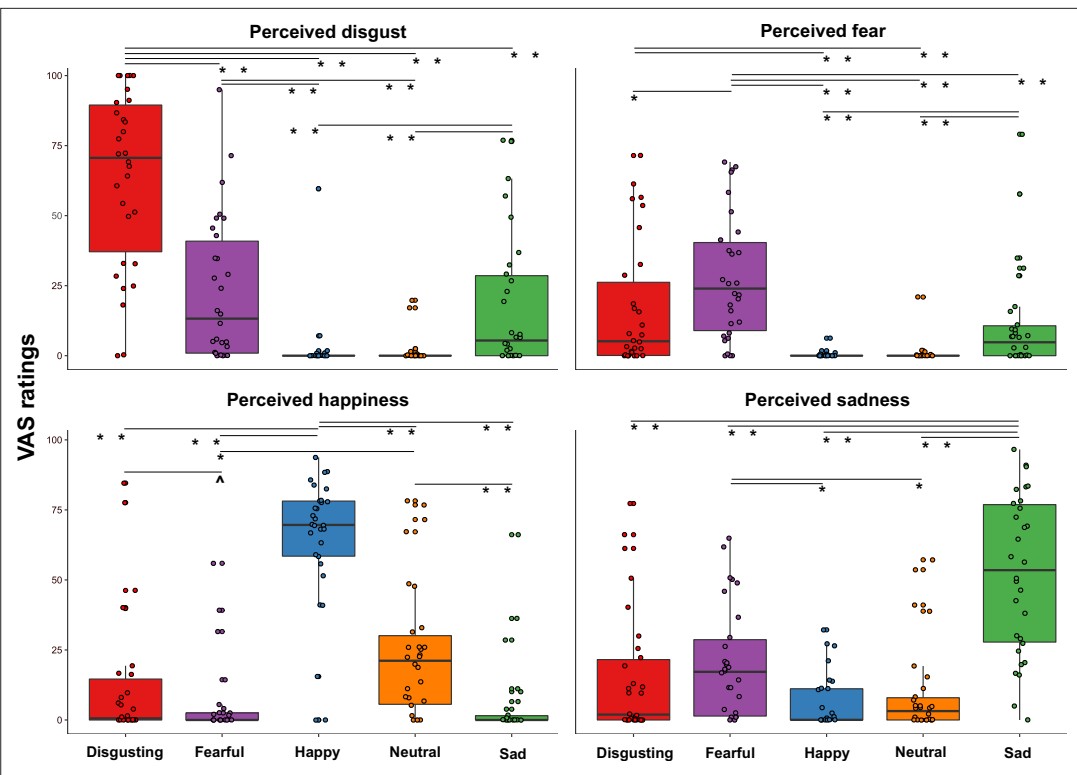

**Figure 2.** Perceived emotions results (pill in the stomach, session 1). Perceived emotions (disgust, fear, happiness, and sadness) measured using 0–100 visuo-analogue scale (VAS) ratings, as a function of the five categories of video clips (disgusting, fearful, happy, sad, and neutral) shown during the first session of this study (i.e., when the capsule was in the stomach). Boxplots are drawn from the first to the third quartiles, with the horizontal lines representing the medians. The lines extending from the top and bottom of the boxplots represent the smallest and largest values within 1.5 times the interquartile ranges (IQR) from the first and third quartiles, respectively. Significant differences refer to Bonferroni-corrected, Wilcoxon matched-pairs tests (threshold for multiple comparisons was set at 0.0125). ∧ p=0.013; * p≤0.01; ** p≤0.001.

clips. Planned post-hoc Bonferroni-corrected Wilcoxon matched-pairs tests showed that *gastric sensations* were higher when participants were asked to observe disgusting and fearful videos. In particular, gastric sensations evoked by disgusting and fearful videos were significantly higher than those evoked by happy video clips (all Zs ≥2.824; all ps ≤0.005). Gastric sensations evoked by disgusting video clips were significantly higher than those evoked by neutral video clips (Z=2.777; p≤0.005), while gastric sensations evoked by fearful video clips were only marginally (considering the Bonferroni corrected significance threshold of 0.0125) higher than those evoked by neutral video clips (Z=2.375; p=0.018). Gastric sensations evoked by sad video clips were significantly higher only with respect to those evoked by happy video clips (Z=2.830; p≤0.005). As to the *cardiac sensations*, we found that all the emotional video clips evoked higher cardiac sensations with respect to the neutral video clips (all Zs ≥3.295; all ps ≤0.001). As for the *respiratory sensations*, we found that they were maximally evoked by disgusting, fearful, and sad videos (i.e. those inducing negative emotions). In particular, respiratory sensations evoked by disgusting, fearful and sad videos were significantly higher than those evoked by happy and neutral video clips, (all Zs ≥3.036; all ps ≤0.002). Happy video clips evoked higher respiratory sensations than neutral video clips (Z=2.515; p≤0.012). Finally, we found that all the emotional video clips evoked higher *feeling of arousal* with respect to the neutral video clips, (all Zs ≥3.295; all ps ≤0.001). See *Figure 3* for a graphical representation of these results.

Due to the fact that results on subjective visceral experience collected in session 2 and session 3 mirrored those of session 1, they are described and plotted in the *Appendix 1—figures 3 and 4*.

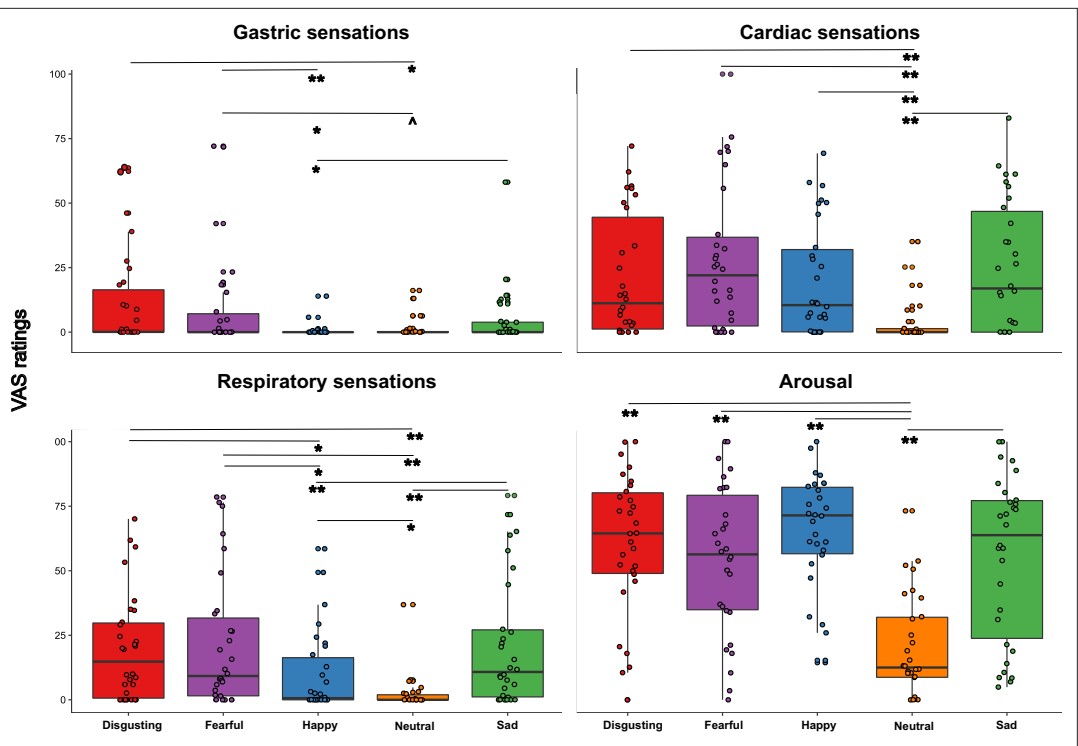

**Figure 3.** Visceral sensations results (pill in the stomach, session 1). Perceived visceral sensations (gastric, cardiac, and respiratory) and arousal, measured using 0–100 visuo-analogue scale (VAS) ratings, as a function of the five categories of video clips: disgusting, fearful, happy, sad, and neutral shown during the first session of this study (i.e. when the capsule was in the stomach). Boxplots are drawn from the first to the third quartiles, with the horizontal lines representing the medians. The lines extending from the top and bottom of the boxplots represent the smallest and largest values within 1.5 times the interquartile ranges (IQR) from the first and third quartiles, respectively. Significant differences refer to Bonferroni-corrected, Wilcoxon matched-pairs tests (threshold for multiple comparisons was set at 0.0125). ∧ p=0.018; * p≤0.01; ** p≤0.001.

## Autonomic and behavioural (oculomotor) markers of perceived emotions: heart rate (HR), heart rate variability (HRV), and spontaneous eye blinks

During the first session, that is when the capsule was in the stomach, we tested changes in heart rate (HR), heart rate variability (HRV), and number of spontaneous eye blinks in response to emotional experiences triggered by five types of video clip (disgust, fear, happiness, sadness, compared to neutral). To do this, after verifying via Kolmogorov-Smirnov tests that all data distributions were normally distributed (all ps >0.05), we conducted three separate repeated-measures ANOVAs with type of video clip as a within-subjects factor and HR, HRV and number of eye blinks as dependent variables.

We found a main effect of the type of video clip on HR (F(4, 112)=5.652, p=0.00035, $\eta$2=0.17) suggesting that participants' HR changed across different emotions. Specifically, Bonferroni-corrected post-hoc analysis showed that HR was higher when participants observed the control neutral scenarios compared to the disgusting and fearful ones (all ps ≤0.008). All the other differences were not significant (all ps ≥0.124), see *Figure 4A*, for a graphical representation of the results.

Additionally, we found a significant main effect of the type of video clip on the RMSSD, (F(4, 112)=2.658, p=0.036, $\eta$2=0.09), suggesting changes in participants' HRV across different emotions. Bonferroni-corrected post-hoc analysis revealed that HRV was higher when participants observed the disgusting video clips compared to the control neutral ones (p=0.049), see *Figure 4B*, for a graphical representation of the results.

Regarding the spontaneous blinks, we found a main effect of the type of video clip (F(4, 108)=12.371, p≤0.0001, $\eta$2=0.32), suggesting that participants' spontaneous eye blinks change across the different

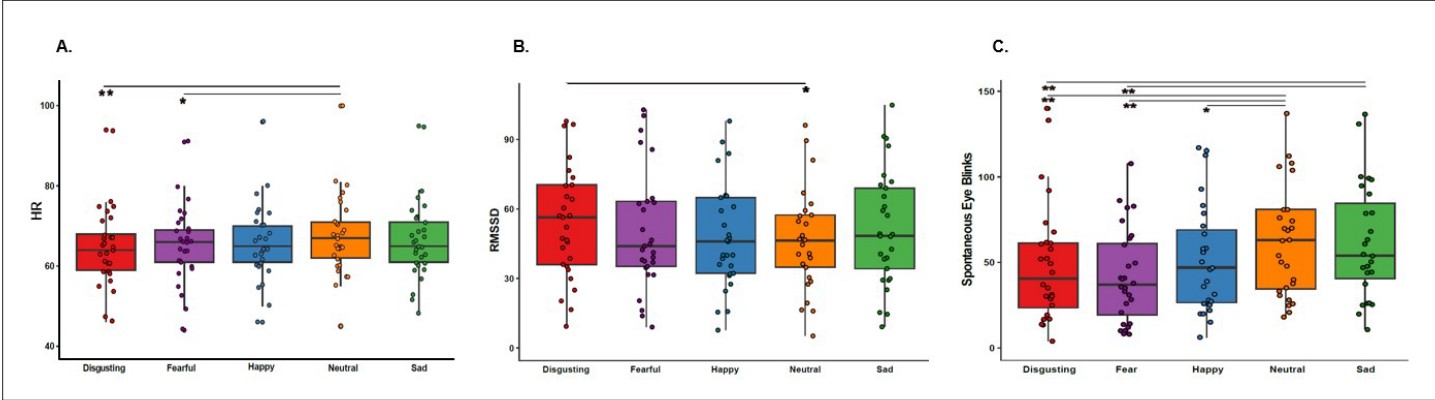

**Figure 4.** Heart Rate (HR), Heart Rate Variability (RMSSD) and Spontaneous Eye Blinks results. (**A**) Shows heart rate, as a function of the five categories of video clips: disgusting, fearful, happy, sad, and neutral shown during the first session of this study (i.e. when the capsule was in the stomach). * p≤0.01; ** p≤0.001. (**B**) Shows heart rate variability, indexed by RMSSD, as a function of the five categories of video clips: disgusting, fearful, happy, sad, and neutral shown during the first session of this study (i.e. when the capsule was in the stomach). * p≤0.05. (**C**) Shows the number of spontaneous eye blinks as a function of the five categories of video clips: disgusting, fearful, happy, sad, and neutral shown during the first session of this study (i.e. when the capsule was in the stomach). * p≤0.05; ** p≤0.01. Boxplots are drawn from the first to the third quartiles, with the horizontal lines representing the medians. The lines extending from the top and bottom of the boxplots represent the smallest and largest values within 1.5 times the interquartile ranges (IQR) from the first and third quartiles, respectively.

emotions. Bonferroni-corrected post-hoc analysis showed that participants blinked less when they observed the disgusting, fearful and happy video clips compared to the control neutral ones (all ps ≤0.009). Furthermore, they blinked more when they observed the sad videos compared to when they observed the disgusting and the fearful ones (all ps ≤0.001), but not when they observed the happy and neutral ones (all ps ≥0.228), see *Figure 4C*, for a graphical representation of the results.

## Gut markers of perceived emotions

### Capsule in the stomach (Session 1): pill-related gastric markers (pH, temperature, pressure)

To test the main hypothesis underlying this study, namely that the emotional experience triggered by the five categories of the video-clips is linked to the inner physiology of the stomach, we conducted a mixed model analysis (Model 1, see *Data analysis* session and Appendix 1 for details) with emotional VAS score as dependent variable, type of video clip and item as independent factors, and stomach pH, temperature and pressure as covariates. Model 1 had a marginal $R^2$=0.51 and a conditional $R^2$=0.60. Visual inspection of the plots did not reveal any obvious deviation from homoscedasticity. Residuals were not normally distributed according to Shapiro-Wilk normality test, but linear models are robust against violations of normality (*Gelman and Hill, 2006*). As for collinearity (tested by means of *vif* function of *car* package), all independent variables had a (GVIF^(1/ (2*Df)))^2<10 except for pressure (GVIF^(1/ (2*Df)))^2=28.858 and pH (GVIF^(1/ (2*Df)))^2=17.986. Type III analysis of variance of Model 1 showed a statistically significant 2-way interaction between item (i.e. perceived disgust, fear, happiness, and sadness) and gastric pH ($F$=8.214, p<0.0001, bootstrap p-value <0.001, Eta2 (partial)=0.05, *Appendix 1—figure 5*), suggesting that the emotional experience reported by participants on the VAS ratings, irrespective of the type of observed video clip, varied according to the pH of the stomach and the type of perceived emotion. Specifically, the follow-up post hoc simple slope analysis, conducted to explore the significant two-way interaction (gastric pH*item), showed that the lower (i.e. more acidic) was the pH of the stomach, the more our participants reported to feel disgusted and afraid, while the higher the pH of their stomach (i.e. less acidic), the more they reported to feel happy. For each 1-unit decrease in stomach pH, a predicted 15.59±5.3 points increase in VAS ratings of disgust was reported ($t$=−2.939, p=0.003). Similarly, for each 1-unit decrease in stomach pH, there was a predicted 10.86±5.3 points increase in VAS ratings of fear ($t$=−2.047, p=0.041). On the other hand, for each 1-unit increase in stomach pH, there was a predicted 21.16±5.3 points increase in VAS ratings of happiness ($t$=3.988, p≤0.001).

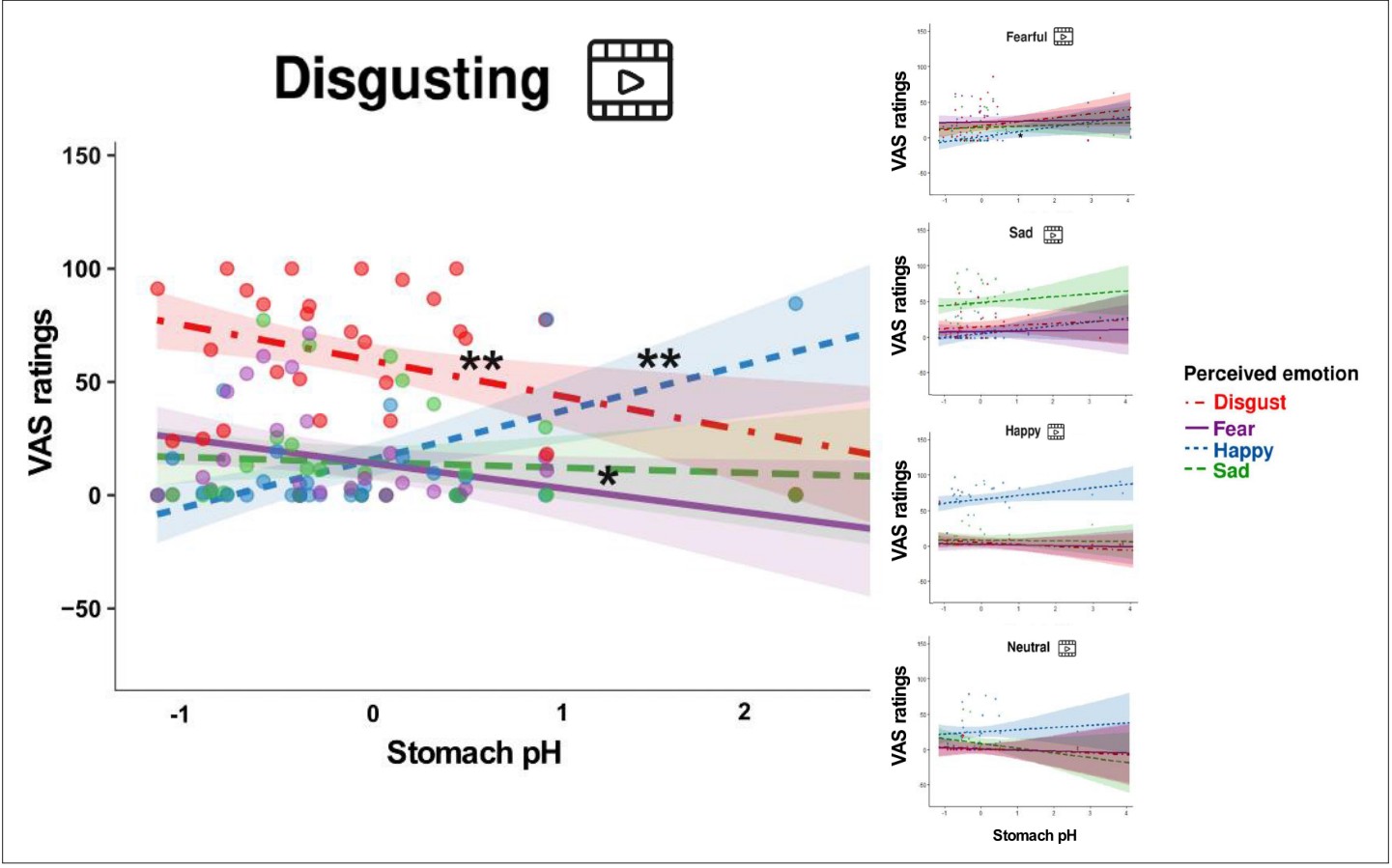

**Figure 5.** Association between the Stomach pH and perceived emotions (three-way interaction between item, stomach pH, and type of video-clip). This figure illustrates the association between stomach pH and perceived emotions (disgust, fear, happiness, and sadness) across the five categories of video clips (disgusting, fearful, happy, neutral, and sad). Notably, it highlights the main differences, primarily observed in the left panel, concerning the disgusting video clips. Panel on the upper right highlights the main differences concerning the fearful video clips. Dots represent participants' VAS ratings following the different types of video clip, shadows indicate Confidence Intervals. The pH has been mean-centred and scaled. *p≤0.05; ** p≤0.01.

The two-way interaction was further defined by a three-way interaction between video-clip content, item, and pH (*F*=1.978, p=0.025, bootstrap p-value = 0.026, Eta2 (partial)=0.05, *Figure 5*), suggesting that emotional experience reported by participants on the VAS ratings varied also as a function of the content of the projected video clips, besides the pH of their stomach and the type of emotion perceived. Follow-up post hoc simple slopes analysis, performed to explain the significant three-way interaction (gastric pH*item*type of video clip), showed that the lower (i.e. more acidic) was the pH of the stomach, the more our participants reported to feel disgusted and afraid, after observing disgusting video clips, while the higher was the pH of their stomach (i.e. less acidic) the more reported to feel happy. For each 1-unit decrease in stomach pH, there was a predicted 15.59±5.3 points increase in VAS ratings of disgust (*t*=−2.939, p=0.003, see the left panel of *Figure 5*). Similarly, for each 1-unit decrease in stomach pH, there was a predicted 10.86±5.3 points increase in VAS ratings of fear (*t*=−2.047, p=0.041, see the left panel of *Figure 5*). On the other hand, for each 1-unit increase in stomach pH, there was a predicted 21.16±5.3 points increase in VAS ratings of happiness (*t*=3.989, p≤0.001, see the left panel of *Figure 5*). Simple slopes analysis also showed that the higher (i.e. more basic) the pH of the stomach, the more our participants reported to feel happy after observing fearful video clips. For each 1-unit increase in stomach pH, there was a predicted 7.51±3.11 points increase in VAS ratings of happiness (*t*=2.412, p≤0.05, see the upper right panel of *Figure 5*). Three-way interactions with stomach pressure and temperature were not statistically significant (all *F*s ≥0.61; all ps ≤0.834). For a detailed description of the additional model results please refer to the Appendix 1, and *Appendix 1—table 1*.

To explore how the emotional induction could modulate the pH of the stomach and how the length of the exposure to that specific emotional induction could also play a role in modulating pH variations, we ran an additional model, Model 2 (see *Data analysis* session and Appendix 1 for details). This model included all the pH datapoints registered using the Smartpill as dependent variable, the type of video clip and the number of the datapoints ('Time') as fixed effects, and the by-subject intercepts as random effects (see Appendix 1 for a detailed description of the model). Model 2 had a marginal $R^2$=0.014 and a conditional $R^2$=0.79. Visual inspection of the plots did reveal some small deviations from homoscedasticity, visual inspection of the residuals did not show important deviations from normality. As for collinearity (tested by means of *vif* function of *car* package), all independent variables had a (GVIF^(1/ (2*Df)))^2<10.

Type III analysis of variance of Model 2 showed a statistically significant main effect of the Time ($F$=20.237, p<0.001, Eta2 <0.01) suggesting that independently from the type of video clip observed, the stomach pH significantly decreased as a function of the time of exposure to the induction.

A significant main effect of the type of video clip was also found ($F$=22.242, p<0.001, Eta2=0.01) suggesting that pH of the stomach changes when participants experienced different types of emotions. In particular, post hoc analysis revealed that pH was more acidic when participants observed disgusting compared to fearful (t=−11.417; p<0.001), happy (t=−15.510; p<0.001) and neutral (t=−3.598; p=0.003) video clips.

Also, pH was more acidic when participants observed fearful compared to happy (t=−4.064; p<0.001), and less acidic compared to neutral (t=7.835; p<0.001) and sad scenarios (t=9.743; p<0.001). Finally, pH was less acidic when participants observed happy compared to neutral (t=11.923; p<0.001) and sad video clips (t=13.806; p<0.001), see *Figure 6*, left panel.

Interestingly, also the double interaction Time X Type of video clip was significant ($F$=3.250, p=0.0113, Eta2 <0.01) suggesting that the time of the exposure to the induction differentially influenced the pH of the stomach depending on to the type of the observed video clip. Simple slope analysis (Bonferroni corrected) showed that while pH did not change over time when observing disgusting (t=−1.2691; p=0.2045) and happy (t=0.4466; p=0.6552) video-clips, it did significantly decrease over time when observing fearful (t=−4.4212; p<0.001), sad (t=−2.0487; p=0.0405) and neutral video clip (t=−2.7956; p=0.0052), see *Figure 6*, right panel.

For a detailed description of Model 2 results please refer to *Appendix 1—tables 3 and 4*.

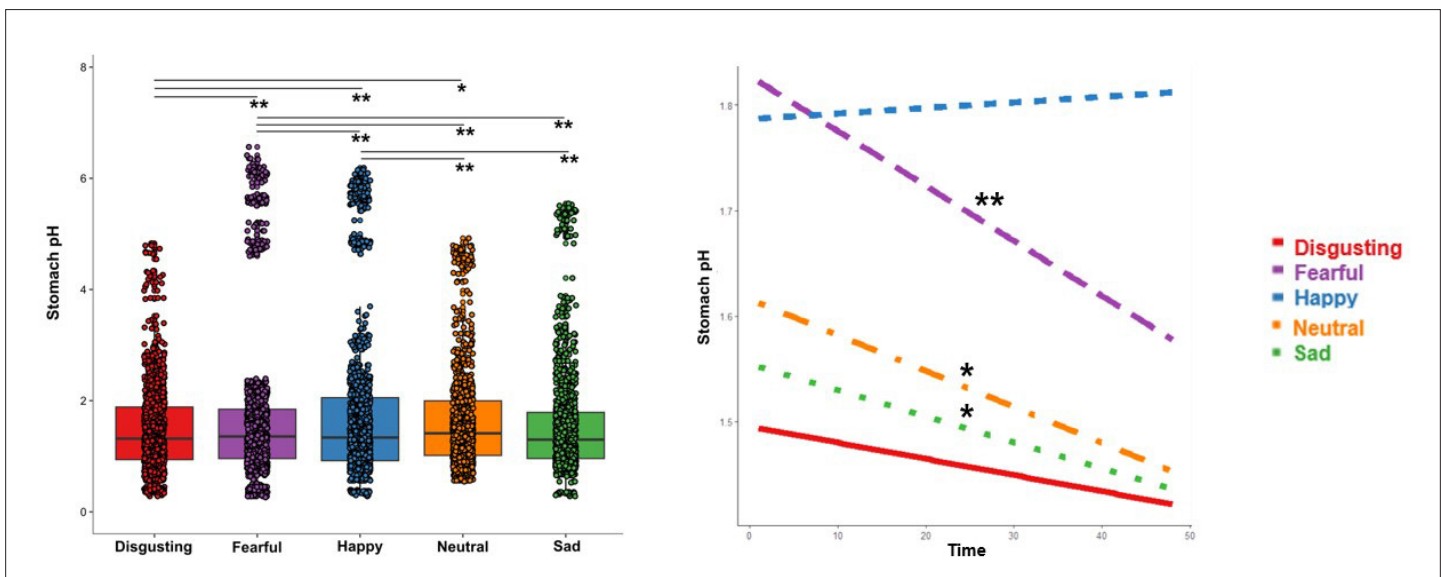

**Figure 6.** Changes of stomach pH as function of type of video clip (left panel) and time, i.e. datapoints, (right panel). *p≤0.05; ** p≤0.001.

## Capsule in the stomach (Session 1): EGG-related gastric markers (normogastric and tachigastric EGG peak frequencies)

With respect to the hypothesis that emotional experience might be linked also to the electromyographic activity of the stomach (EGG) we ran Model 3 and 4 (see above, Data analysis paragraph and Appendix 1 for details). These models contained the emotional VAS ratings as dependent variables, the type of video clip and items as independent factors, the individual normogastric and tachygastric EGG peak frequency respectively as covariates and the by-subject intercepts as random effects.

Around 10% of the EGG data were missed due to technical reasons or noise in the signal. Model 3 had a marginal $R^2=0.48$ and a conditional $R^2=0.56$. Visual inspection of the plots did not reveal any obvious deviation from homoscedasticity. Residuals were not normally distributed according to Shapiro-Wilk normality test, but linear models are robust against violations of normality (*Gelman and Hill, 2006*). As for collinearity (tested by means of *vif* function of *car* package), all independent variables had a (GVIF^(1/ (2*Df)))^2<10. Type III analysis of variance of Model 3 showed a statistically significant two-way interactions between video-clip content and item ($F=35.49$, $p<0.001$); however, the three-way interaction between video-clip content, item, and EGG peak frequency was not statistically significant ($F=0.57$, $p=0.868$), suggesting that emotional experience reported by participants on the VAS ratings was only influenced by the content of the projected video clips, but not by the EGG peak frequency of their stomach extracted from the normogastric band. For a detailed description of model 3 results please refer to *Appendix 1—table 5*. Model 4 mirrored results of Model 3 with no significant three-way interaction, suggesting that emotional experience reported by participants on the VAS ratings was not influenced by the EGG peak frequency of their stomach extracted from the tachigastric band.

Finally, we explored whether normogastric and/or tachygastric cycle changed in response to specific emotional experience. After checking that normogastric and tachygastric peak frequencies were normally distributed (all ps >0.05), we ran two separate ANOVAs on the individual peak frequencies in the normogastric and tachygastric range. Each analysis had the type of video clip as within-subjects factor. The ANOVA performed on the normogastric rhythm was not significant (F(4, 44)=1.037, p=0.399) suggesting that the gastric rhythm did not change while participants observed the different emotional video clips. In contrast, the ANOVA performed on the tachygastric rhythm did show a significant main effect (F(4, 112)=2.907, p=0.025, Eta2 (partial)=0.09). However, the only comparison that survived the Bonferroni correction was the one between happy and fearful video clips, namely participants' tachygastric cycle was faster when they observed happy vs fearful video clips (p=0.038) see *Appendix 1—figure 6* for a graphical representation of the results.

## Capsule in the small (Session 2) and large bowel (Session 3)

To test the more exploratory hypothesis that the emotional experience triggered by the five categories of the video-clips is linked to the inner physiology of the small and large bowel, we conducted two separate mixed model analysis (Model 5 and Model 6, see Appendix 1 for a detailed description) with VAS ratings as dependent variable, type of video clip and item as independent factors, and small and large bowel pH, temperature and pressure respectively as covariates.

Model 5 (see below, *Data analysis* procedure) did not report any significant two- and three-way interactions between video-clip content, items and small bowel pH, pressure, and temperature, suggesting that the emotional experience reported by participants on the VAS was only influenced by the content of the projected video clips, but not by pH, pressure, or temperature recorded in the small bowel. For a detailed report of the model results please refer to *Appendix 1—table 6*.

Similarly, Model 6 (see below, *Data analysis* procedure) did not report any significant two- and three-way interactions between video-clip content, items and small bowel pH, pressure, and temperature, suggesting that the emotional experience reported by participants on the VAS was only influenced by the content of the projected video clips, but not by pH, pressure, or temperature recorded in the large bowel. For a detailed report of the model results please refer to *Appendix 1—table 7*.

## Discussion

To explore the direct link between GI physiology and emotional experiences, we asked healthy male participants to observe a series of validated video clips (*Tettamanti et al., 2012*) while electrical,

chemical, and thermal signals originating from their GI system were recorded. Specifically, we induced disgust, fear, sadness, and happiness three times, when an inert ingestible pill that gauged pH, temperature and pressure was in the stomach, small and large bowel. When the pill was in the stomach we also recorded, via electrogastrography (EGG), participants' gastric myoelectric activity.

In addition to the objective GI measures, we collected participants' subjective perception evoked by the different stimuli. Specifically, at the end of each emotional block, we asked participants to evaluate by means of 0–100 visuo-analogue scales (VAS) their perceived emotions, visceral sensations (gastric, cardiac, and respiratory) and arousal.

Finally, to complement those measures with more consolidated ones, we also collected changes in heart rate (HR), heart rate variability (HRV), and spontaneous eye blinks as non-gastric behavioral and autonomic markers of the emotional experience.

In keeping with previous findings (*Tettamanti et al., 2012*), we found that the four emotional categories of the video clips were clearly able to induce all the intended emotions (*Figure 2*). The effect was present in all the three experimental sessions, indicating that emotional induction was successful and no habituation over time occurred. It is also worth noting that the four types of emotional video clips differed from the neutral ones which were perceived as less arousing (*Figure 3*, bottom right panel). These findings are consistent with previous literature suggesting that perceived arousal is indeed an essential component of the emotional experience (*Posner et al., 2005*; *Russell and Barrett, 1999*) even if, unlike previous findings (*Vianna et al., 2006*; *Vianna and Tranel, 2006*), we did not find higher ratings of arousal triggered by negative vs. positive video clips suggesting that the reported effects are genuinely associated with emotional experience in general.

Our results on the visceral sensations suggest that the subjective perception of different bodily signals, particularly of gastric feelings, varied across the different emotions. Specifically, gastric sensations were higher after observing fearful videos and maximal after observing disgusting videos, compared to the other emotional video clips (*Figure 3*, upper left panel). Respiratory sensations, in turn, were higher after observation of disgusting, fearful and sad video clips, compared to the happy and neutral ones (*Figure 3*, bottom left panel). Cardiac sensations instead did not differ between the emotional categories and were higher with respect to non-emotional stimuli (*Figure 3*, upper right panel). These results confirmed the idea that interoception (i.e. the awareness of physiological changes coming from internal organs) is a key component of emotional feelings. This is in line with theories (*Barrett, 2017*; *Barrett, 2014*) that suggest a key role of bodily signals in the experience of emotions. However, the results also suggest that specific visceral signals uniquely characterize each emotional state, indicating that conceiving distinct emotional experience (in line with somatic theories of emotions *Damasio, 1999*; *Harrison et al., 2010*; *James, 1994*) provides a clearer description of our data. With respect to our main hypothesis – that internally and externally recorded GI markers are linked to emotions – we found that gastric physiology does play a significant role in emotional experiences. Specifically, the more acidic the pH recorded in participants' stomach, the more they reported disgust and fear; the less acidic the gastric environment, the more participants reported happiness, independently from the observed category of video clips (*Appendix 1—figure 5*). Moreover, when participants were exposed to disgusting video clips, more acidic gastric pH was associated with higher reports of disgust and fear, while less acidic gastric pH was associated with higher reports of happiness. Similarly, when participants were exposed to fearful video clips, the less acidic was their pH, the more they reported to perceive happiness (*Figure 5*). Overall, these results were only found for the stomach and did not extend to the (small and large) bowel. Although it is true that the order was fixed, so that intestine data were always recorded after participants had already been exposed to a first round of video clips when the capsule was in the stomach, at the subjective level (i.e. emotional and visceral VAS ratings) emotional stimuli did induce the same experience in all the three sessions. In the light of this evidence and of the literature, it seems that stomach physiology, and not a pure novelty effect triggered by the emotional stimuli, plays a specific and crucial role in subjective emotional experience.

To complement results showing a key role of the stomach pH in the perceived emotional experience, we also tested the hypothesis that independently from the conscious emotional experience (i.e. emotional VAS ratings) the pH varies during the observation of the different types of video clips. In fact, we found that stomach pH was generally more acidic when participants observed disgusting video clips and generally less acidic when participants observed happy video-clips. These effects did not

change over time. Differently, we found that stomach pH was more acidic when participants observed fearful compared to happy video-clips, and less acidic compared to neutral and sad scenarios, with a general decrease over time of the pH, see *Figure 6*.

Our findings about the link between pH acidity of the stomach and (perceived) emotions (disgust, fear, and happiness) are in line with the anecdotal reports described by *Beaumont, 1833* about Alexis St. Martin. He was a Canadian fur trader with a gunshot-created permanent fistula, through which Beaumont could access secretion samples from the patient's stomach and report that when he was exposed to emotional experiences, mainly stress and anger, his gastric secretion changed color. Following these first inspiring but qualitative findings, very few studies explored changes in the human gastric milieu during higher-order cognitive and emotional processes. To the best of our knowledge, two single case studies conducted in the past century reported changes at the level of gastric acid output following anxiety induction procedure (*Bennett and Venables, 1920*) and a decrease in stomach motility and acid secretion during depression (*Wolf and Wolff, 1947*). Only after more than four decades, a somewhat systematic investigation showed that an acute psychological stress (i.e. caused by performing mental arithmetic and solving anagrams) not only increased arterial blood pressure and heart rate, but also stimulated gastric acid output recorded by means of a naso-gastric tube in healthy participants scoring high in impulsivity traits (*Holtmann et al., 1990*). Although in the present study we did not measure gastric secretion, which is a procedure that usually relies on invasive techniques (*Ghosh et al., 2011*), we show that stomach pH might be an important marker of emotional experience and that ingestible pills could be a non-invasive and sensitive tool to measure it.

Regarding the EGG data, we did not find any association between the perceived emotions (i.e. emotional VAS ratings) and individuals' peak frequency of the normogastric and tachigastric band, differently to what previous literature suggested (*Harrison et al., 2010*; *Shenhav and Mendes, 2014*, *Gianaros et al., 2001*; *Peyrot des Gachons et al., 2011*; *Shenhav and Mendes, 2014*; *Stern et al., 2001*; *Vianna and Tranel, 2006*). However, the emotional induction did influence the EGG tachigastric peak frequency. Post hoc analyses showed that during the happy videos the gastric cycle was faster compared to fearful video clips. Future studies with longer emotional induction procedures, more sophisticated analyses, and multiple-channel montage are needed to better investigate if and how emotional video clips might affect over time the whole spectrum of gastric frequencies and clarify the underlying mechanisms.

Unlike stomach pH, neither GI pressure nor temperature were associated with the emotional experience in the present study. It is important to underline that SmartPill capsules record temperature with a relatively low sample rate (every 20 s). Therefore, it is possible that different technologies with higher sample rates might be more sensitive to detect changes of endoluminal temperature triggered by the emotional experience.

Complementing the results found in the deep gastric realm, we also found that disgusting stimuli induced a significant increase in heart rate variability (HRV), as indexed by an increase in the root mean square of successive beat-to-beat interval differences (RMSSD), compared to the neutral scenarios and that, together with fearful ones, induced a decrease in heart rate (HR). The results concerning fearful video clips contradict several pieces of evidence published in literature reporting an increase of HR (e.g. *Aue et al., 2007*; *Kreibig et al., 2007*). However, they align with a few studies (*Fredrickson and Levenson, 1998*) that have instead documented a decrease in heart rate in response to video clips eliciting fear of falling (which was also present in the fearful video clips we used). With respect to the results on disgust, they align with evidence published in literature reporting a decrease in HR (e.g., *Baldaro et al., 2001*; *Meissner et al., 2011*; *Palomba et al., 2000*; *Rohrmann and Hopp, 2008*) and an increase in HRV (e.g., *Rohrmann and Hopp, 2008*) suggesting that during induction of disgust there is a sympathetic-parasympathetic coactivation with eventually a heightened parasympathetic activity (e.g. *Brownley et al., 2000*). It is also important to consider that disgust is not a unidimensional construct, and that the different results found in the literature about the role of autonomic nervous system might depend on the stimuli used to induced it (e.g. core disgust stimuli; blood/body boundary violation disgust stimuli or moral disgust stimuli), see for example *Ottaviani et al., 2013* and *Shenhav and Mendes, 2014*.

Finally, we also found a general decrease of the spontaneous eye blinks during the observation of both positive and negative emotions compared to neutral scenarios with the only exception of the sad video clips. These findings support the idea that besides hydrating the eyes and protecting them

against foreign objects (e.g. *Evinger et al., 1991b*; *Evinger et al., 1991a*) blinking is a physiological mechanism that regulates information loss. It is linked to cognitive functions, particularly those related to attention and vigilance (e.g. *McIntire et al., 2014*), and thus might be conceived as an index of the salience triggered by the stimuli.

Overall, and in line with theoretical and empirical evidence (*Damasio, 1999*; *Harrison et al., 2010*; *James, 1994*; *Lettieri et al., 2019*; *Stephens et al., 2010*), our findings suggest that specific patterns of subjective, behavioural, and physiological measures are linked to unique emotional states.

We acknowledge that our results, although novel, are restricted to a sample of male participants, and more importantly they need to be replicated. We also acknowledge that future studies should better investigate the mechanisms underlying the role of the pH in the emergence of specific emotion. For instance, pharmacologically manipulating stomach pH during emotional induction, not only for basic emotions but also for exploring complex emotions such as moral disgust (*Rozin et al., 2009*), would enable researchers to generalize these findings and examine the directionality of this relationship. In particular, it would be interesting to see how emotions are experienced when classic anti-acids or proton pump inhibitors are administered or to see what happens to emotional experience after normalization of the gastric rhythm when an *anti-emetic* and a prokinetic agent, such as domperidone, is administered (*Nord et al., 2021*).

Delving into the neural correlates underlying these phenomena, specifically focusing on the bidirectional pathways linking the stomach and the brain would also be of significant interest. For instance, investigating the potential role of brain regions like the anterior cingulate cortex (ACC) and the insula (AI), conceived as primary interoceptive cortices with functions encompassing homeostatic, emotional, limbic, and sensorimotor processes, would be valuable. Moreover, considering regions such as the nucleus tractus solitarii (NTS) in the brainstem, which serves as the pathway for major inputs projecting information to higher-order brain areas, could provide crucial insights (see the recent systematic review on the neural networks that underpin nausea in humans by *Varangot-Reille et al., 2023*). Additionally, it will be pivotal to map differences in stomach-brain coupling activity (*Rebollo et al., 2018*; *Rebollo and Tallon-Baudry, 2022*; *Richter et al., 2017*) during different emotional experiences.

We believe that the present findings suggest that the stomach plays a fundamental role in emotional experience and have the potential to open new avenues for studying the unexplored influences of the neurobiology of the GI system on typical and atypical emotional processes. Our approach may be adopted for example when studying the contribution of the gastric signals in people showing dysfunctional emotional processing like autism spectrum conditions and depression. Evidence shows that the above-mentioned conditions are characterized by comorbidities with GI problems (people with irritable bowel syndrome [IBS] are much more likely to develop depression than healthy controls *Shah et al., 2014*, and autistic persons are more likely to develop IBS compared to controls *Kim et al., 2022*). Finally, our approach can be also adopted for studying emotional processes in conditions characterized by alterations of the GI physiology itself and its awareness, like persons with eating disorders (for a recent review see *Khalsa et al., 2022*).

## Materials and methods
### Participants
To avoid the confounding effects played by sex both at level of emotional processing/experience (e.g. *Kret and De Gelder, 2012*) and of gastric physiology (e.g. *Tolj et al., 2007*), we choose to recruit only male participants to test a more homogeneous sample. Also, considering sex differences within emotional experience, testing men constitutes a conservative approach. We are well aware, though, that studies in women are necessary to generalize these findings and we are already working on this aspect.

Here, 31 healthy male participants (age: M=24.42, S.D.=2.8, range = 20–30 years) were recruited via the Social and Cognitive Neuroscience Laboratory volunteer database and through personal contacts to take part in this study. A preliminary structured questionnaire ensured they all were eligible to the experimental procedure and to the SmartPill (SmartPill Motility Testing System, Medtronic plc) capsule ingestion (see the SmartPill system paragraph for a detailed description of this device). Exclusion criteria included diagnosis of psychiatric, neurological, or swallowing disorders; gastric bezoars;

history of any abdominal/pelvic surgery within the previous three months; suspected or known strictures, fistulas, or physiological/mechanical obstruction within the GI tract; dysphagia to food or pills; Crohn's disease or diverticulitis; body mass index ≥40; age <18 years; cardiac pacemakers or defibrillators; assumption of any medication or substance that could interfere with pH values and GI motility (*Saad, 2016*), such as tobacco (within 8 hr prior to capsule ingestion), antacids and alcohol (within 24 hr), laxatives (within 48 hr); antihistamines, prokinetics, antiemetics, anticholinergics, antidiarrheals, narcotic analgesics, non-steroidal anti-inflammatories (within 72 hr), or proton pump inhibitors (within seven days). The protocol of this study was approved by the local Institutional Review Board (Fondazione Santa Lucia ethics committee, Protocol CE/PROG.636) and was performed in accordance with the Declaration of Helsinki (1991). All participants were naïve to the purpose of the experiment, provided their written informed consent before starting the experimental procedure, and received a reimbursement for their participation at the end of the study. The same day of the experiment, they also took part in two additional studies, each with different aims and different hypotheses.

## SmartPill system

The SmartPill system (SmartPill Motility Testing System, Medtronic plc) was originally approved by the US Food and Drug Administration for the evaluation of suspected gastroparesis and colonic transit in patients with chronic idiopathic constipation (*Aburub et al., 2018*). In a position paper, the American

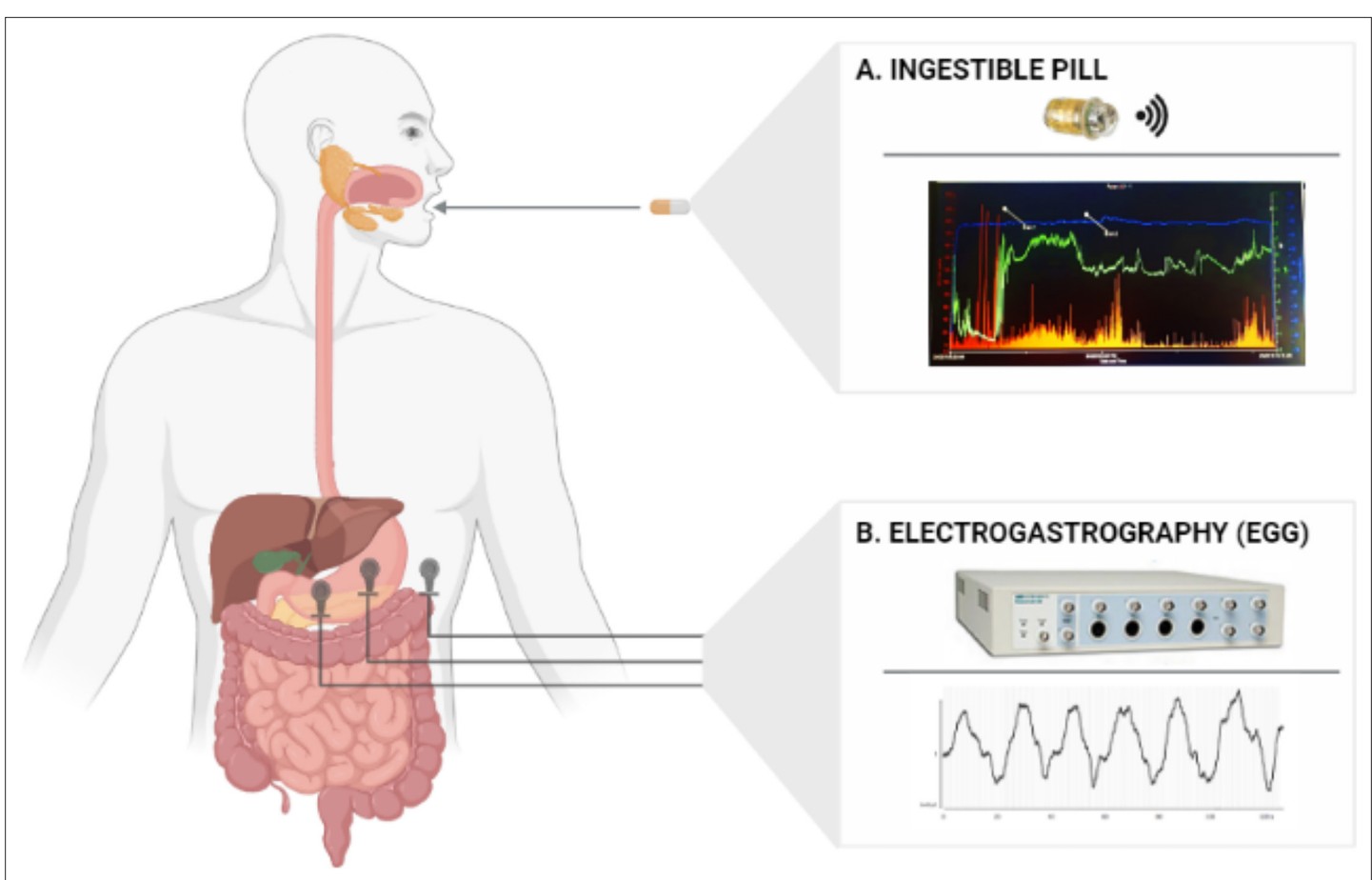

**Figure 7.** Gastro-intestinal (GI) markers collected in the present study. (**A**) Shows the SmartPill capsule used to record temperature, pH, and pressure of the gut and the intestine as well as a typical graph created by the dedicated MotiliGI software, plotting temperature (blue line), pH (green line), and pressure (orange bars) across the entire the GI tract. (**B**) Shows the device (PowerLab, ADInstruments Ltd) used to record electrogastrography (EGG) and a typical gastric myoelectric activity of a participant. The 1-channel bipolar electrode montage for the EGG recording is displayed on the left side of the figure.

and European Neurogastroenterology and Motility Societies indicated that this system is useful in the assessment of clinical disorders affecting the GI tract (*Rao et al., 2011*). The SmartPill system consists of a wireless, single-use, ingestible capsule; a receiver; a docking station; and software (i.e. MotiliGI) installed on a dedicated laptop (*Figure 7A*). The ingestible pill is a cylindrical polyurethane capsule, 26.8 mm long, 11.7 mm wide, and it weighs 4.5 g (*Hasler, 2014*). It houses a pH sensor with a range of 1–9 units (accuracy of ±0.5 units); a solid state temperature sensor with a range of 20–42°C (accuracy of ± 1 °C), a solid-state pressure sensor with a range of 0–350 mmHg (accuracy of ± 5 mmHg for values <100 mmHg and accuracy of ±10% for values >100 mmHg), two silver oxide batteries (duration >5 days), and a transmitter (broadcast frequency: 434.2 MHz). Prior to ingestion, the capsule is activated through a magnetic fixture and the pH sensor is calibrated through a buffer solution. After ingestion, the capsule starts recording intraluminal pressure, pH, and temperature and sending these data to the external radio receiver that participants wear around their waist. For the first 24 hr, temperature data are transmitted every 20 s, pressure every 0.5 s, and pH every 5 s; after the first day, sampling frequencies are halved. By means of the MotiliGI software (Medtronic), pH, temperature, and pressure data can be graphically displayed in real time or reviewed after the pill expulsion. Combining pH and temperature data, MotiliGI identifies the specific region of the GI tract in which the pill is located. The arrival of the capsule in the stomach is indicated by a rapid rise in temperature from ambient to body temperature and by a drop in pH (reflecting passage into the acidic environment of the stomach). The arrival of the capsule in the small intestine through the pylorus is instead indicated by an abrupt increase of ≥2 pH units from the gastric baseline. Likewise, a subsequent gradual decrease of ≥1 pH unit for at least 10 consecutive minutes is considered as a sign that the pill left the small intestine and entered the large intestine. If the pH decrease cannot be observed, the MotiliGI software relies on pressure data to mark the transition between small and large bowel. The expulsion of the capsule is defined by a drastic drop in temperature followed by loss in recorded signal after the subject defecated. In our sample, 30 out of 31 subjects displayed a prototypical pattern of pH increase and decrease, thus indicating when the pill was in each of the three areas of interest. For the remaining subject, the software was still able to localize the GI regions that the pill went through based on temperature, pressure, and time data.

## Electrogastrography (EGG), Electrocardiography (ECG), and ocular behaviour (blinks)

During the first session of this study (see the *Experimental procedure* paragraph for a detailed description), besides GI pH, temperature, and pressure, we also recorded participants' surface gastric myoelectric activity by placing three cutaneous electrodes on their abdominal skin over the stomach (*Figure 7B*) and ocular behavior, namely spontaneous eye blinks. Electrogastrography (EGG) is a non-invasive technique that allows the recording of (i) the slow (~0.05 Hz) electrical gastric rhythm, constantly generated in the stomach wall within the so-called 'dominant pacemaker' area located in the greater curvature of the mid/upper corpus; (ii) the more transient smooth muscle activity inducing gastric peristaltic contractions (see *Wolpert et al., 2020* for an extensive review). Here, we used a 1-channel EGG bipolar montage that consists of three pre-gelled disposable Ag/AgCl electrodes placed in a standardized position (*Yin and Chen, 2013*) and attached to a PowerLab data acquisition device (ADInstrument Ltd). Specifically, the first recording electrode was placed halfway between participants' xiphoid and their umbilicus, the second 5 cm up and 5 cm to the left of the first (taking the left side of participants as a reference), and the ground electrode was placed on the left costal margin (see the left part of *Figure 7*). By placing the three electrodes on the abdomen of the participants, we also acquire participants' electrocardiogram (i.e. ECG) that we used to specifically calculate heart rate (HR) and heart variability measures (HRV). For the HRV we focused on an index of variability in the domain of time, the root mean square of successive beat-to-beat interval differences (RMSSD, ms), since we did not have a long period of ECG registration. The RMSSD is considered a measure of vagal regulation of HR (*Shaffer and Ginsberg, 2017*). To do that firstly filtered the raw data using a band pass filter (0.1–30 Hz), we visually inspected them, finally we imported them on Kubios HRV software (*Tarvainen et al., 2014*). Artifacts and ectopic beats were corrected using a threshold-based correction, R peaks were automatically detected and HR and RMSSD measures were extracted for each participant and each experimental condition. Finally, by placing two bipolar electrodes above and below the right eye, vertical electro-oculograms (EOGs) were recorded. Number of spontaneous

blink movements were calculated for each participant and each condition using LabChart 7 software (AD Instruments, Australia).

## Emotional induction task

An illustration of the emotional induction procedure adopted in this study is provided in *Figure 1*. Participants were asked to observe four blocks of a series of emotional video clips with different emotional content. Each block aimed at triggering a specific basic emotion, namely disgust, fear, happiness, and sadness. In addition to the four blocks, participants also observed one block containing a series of neutral video clips (control condition). The five blocks were presented in randomized order across participants and contained 24 emotional video clips each. Video clips were edited color soundless film excerpts lasting ~9 s each. The order of video clips within each block was randomized. All the 120 video clips were selected and validated by Tettamanti and collaborators (*Tettamanti et al., 2012*). Clips showing mutilations or contamination scenarios (e.g. getting in touch with offensive or infective agents) triggered disgust; clips portraying human or animal aggressions and dangerous situations (e.g. falling or drowning) elicited fear; clips depicting amusement scenes, significant achievements or meaningful human interactions (e.g. a meeting between lovers) triggered happiness; clips containing scenarios of death, loss or sickness, (e.g. someone holding their beloved dead) triggered sadness; finally, clips with scenarios of routine human actions and activities (e.g. working or speaking) did not contain any specific emotion and therefore were considered neutral. For a detailed description of the stimuli (and their validation), please refer to the original study (*Tettamanti et al., 2012*) and to the *Appendix 1—table 8*, in which a brief description of the content of each of the video clips employed in our study is provided. At the end of each block, participants answered an 8-item questionnaire relative to their emotional and visceral feelings induced by the observation of the video clips. Four items measured the *emotional experience* in terms of perceived intensity of each emotion, three additional items measured the *visceral experience* in terms of perceived intensity of each visceral feeling and one item measured the perceived *arousal*. Items were presented in a randomized order and participants provided their responses by clicking with the mouse on separate 0 ('Not at all') - 100 ('Very much') visuo-analogue scales (VAS). To avoid carryover effects induced by each category of the emotional video clips, blocks were interspersed with two-minutes washout pauses in which participants simply rested. Instructions, stimuli presentation and collection of responses concerning the emotional induction task were handled by a custom MATLAB algorithm. *Table 1* shows the complete list of the questionnaire items following each block of video clips.

**Table 1.** Eight-item questionnaire.

English translation and original Italian text of the 8-items questionnaire probing the participants' emotions, visceral feelings, and arousal after each block of video clips. Ratings were provided along 0–100 VAS scales.

| Construct | Category | Item (English) | Item (Original language: Italian) |
|---|---|---|---|
| Emotional experience | Perceived disgust | How much disgust did you feel? | Quanto disgusto hai provato? |
| Emotional experience | Perceived fear | How much fear did you feel? | Quanta paura hai provato? |
| Emotional experience | Perceived happiness | How much happiness did you feel? | Quanta felicità hai provato? |
| Emotional experience | Perceived sadness | How much sadness did you feel? | Quanta tristezza hai provato? |
| Visceral experience | Perceived gastric sensations | Did you feel retching, stomach contractions, stomach cramps or other gastro-intestinal sensations? | Hai provato conati di vomito, contrazioni gastriche, crampi allo stomaco o altre sensazioni gastrointestinali? |
| Visceral experience | Perceived cardiac sensations | Did you feel your heart change (i.e. accelerate/slow down or beat)? | Hai sentito il tuo cuore cambiare (accelerare/rallentare/palpitare)? |
| Visceral experience | Perceived breathing sensations | Did you feel your breath change (heavy breath / holding breath/lack of air?) | Hai sentito il tuo respiro cambiare (respirare affannosamente/trattenere il respiro/mancanza d'aria)? |
| Arousal | Perceived arousal | Did watching the clips bored/aroused you? | Vedere i video ti ha annoiato/coinvolto? |

## Experimental procedure

Testing sessions began with an overnight fast, the avoidance of tobacco and alcohol, and the discontinuation of medications potentially altering gastric pH and GI motility (see above). Participants read, filled, and signed the informed consent form. After verifying that all the requirements for the participation were met, participants ate a standardized ~260 kcal breakfast consisting of egg whites (120 g), two slices of bread and jam (30 g) to make sure that GI transit times of the SmartPill capsule were not affected by meal variability. Meanwhile, in a dedicated room, we activated the capsule through a magnetic fixture and calibrated the capsule pH sensor (see Materials and methods above). After pH calibration was complete, the pill started transmitting data to the radio receiver. PH calibrations were performed correctly in all but one participant, therefore pH data from one participant were not considered in the statistical analyses since they were not reliable. Thus pH data were available for 30 out of 31 participants. Data recorded through the capsule came with a relative timestamp indicating the number of seconds elapsed from calibration. Since SmartPill data were not associated with an absolute time frame, we synchronized each calibration with an external clock that provided us with the required absolute time frame (see *Data analysis* session for a detailed description of the data pre-processing). At that point, participants swallowed the SmartPill capsule while drinking a glass of water (120 ml). A medical doctor supervised the ingestion procedure to help in case of swallowing problems. All participants ingested the pill without any trouble. After the ingestion, they fastened the receiver around their belt and lay supine on a deck chair. The experimenter cleaned the skin corresponding to the position of the EGG electrodes and afterwards attached them according to a 1-channel bipolar montage (*Yin and Chen, 2013*; see *Figure 7*). After checking that a highly acidic pH (~1–2) was recorded by the capsule, indicating that it reached the stomach, a 15-min resting-state SmartPill capsule/EGG baseline session started. During this 15-min resting-state session, participants were left alone in the room and instructed to relax while keeping their eyes open. At the end of the baseline session, participants underwent the emotional task during which they observed five blocks of film excerpts while stomach pH, temperature, pressure, and EGG signal were recorded. After each block, participants answered the 8-item questionnaire relative to their subjective emotional and visceral experiences triggered by the video clips. Participants repeated the emotional task two more times, namely when the SmartPill capsule reached the small intestine (usually after 2–5 hr from the capsule ingestion) and when it reached the large intestine (usually after 2–6 hr from the stomach-small bowel transition). After around 6 hr from the beginning of the experimental session, participants were provided with a meal and ad libitum water. After 8 hr, they were also allowed to smoke. Between the experimental sessions they were free to work, study, or rest, although they were asked to avoid strenuous physical exercise and alcohol consumption. At the end of the last experimental session, that is at the end of the emotional task and while the capsule was in the large bowel, participants could leave the laboratory. They were asked to keep the receiver with them until they noticed that after defecation data transmission stopped due to pill expulsion, an event that ordinarily happens after 10–59 hr from ingestion (*Saad, 2016*). After the expulsion, participants came back in the lab to return the receiver.

## Data analysis

Details on SmartPill and EGG data pre-processing can be found in the Appendix 1.

## Statistical data analyses

To exclude habituation effects, we tested whether the five blocks of video clips triggered different subjective emotional and visceral experiences in our participants in all the three experimental sessions. To do so, separate Friedman ANOVAs followed by Bonferroni-corrected Wilcoxon matched-pairs tests (p≤0.0125 was considered as significance threshold) were performed on the VAS ratings relative to the emotional and visceral experience triggered by the different types of video clip. A non-parametrical approach was used because in several conditions VAS ratings were not normally distributed according to the Kolmogorov–Smirnov (K-S) tests and Skewness and Kurtosis z-scores, as suggested by *Field, 2009*. During the first session, that is when the capsule was in the stomach, we also tested changes in heart rate (HR), heart rate variability (HRV), and number of spontaneous eye blinks in response to emotional experiences triggered by five types of video clip (disgust, fear, happiness, sadness, compared to neutral). To do this, after verifying via Kolmogorov-Smirnov tests that all data distributions were normally distributed (all ps >0.05), we conducted three separate repeated-measures

ANOVAs with type of video clip as a within-subjects factor and HR, HRV and individual HR, RMSSD and number of eye blinks respectively as dependent variables. Post-hocs with Bonferroni correction was adopted when significant main effects were found.

Then, we estimated the impact of GI pH, temperature, and pressure (as measured by the Smart-Pill capsule) on the emotional experience (as reflected in the VAS ratings). To do so, three separate mixed-effects models (Model 1; Model 5; and Model 6) were run, one for each GI region (stomach, small bowel, and large bowel) having as dependent variable the VAS ratings relative to the emotional experience questionnaire, and as fixed factors *emotion* (induced by the different type of the video clip: disgust, fear, happy, neutral, and sad) and *item* (the type of question participants were required to answer: perceived disgust, fear, happiness and sadness) as fixed factors, *pH, temperature and pressure (mean-centred and scaled)* as continuous fixed factors, and by-subject intercepts as random effects. Significant main or interaction effects were replicated if the random slope of the *emotion* was included as a random effect in the mixed models. Moreover, to reduce the probability to describe false positives due to multiple comparisons, we used the *emtrends* function of the *emmeans* package (**Lenth, 2019**) to perform post hoc tests on the more conservative model (the one with the condition as random slope) with Bonferroni-adjusted p values (see **Appendix 1—table 2** for the detailed results). Due to the fact that models with random slopes yielded a boundary (singular) fit we decided to report the results coming from the less complex, intercept-only models in the main text. Finally, to better explore the role of the pH on emotional experience, and check variations of pH in function of the type of video clip and time, we ran an additional model, Model 2. This time, to increase the power of the analysis, we considered all the datapoints collected by using the Smartpills (not only the average pH per condition) regardless of the participants' emotional experience (i.e. emotional VAS ratings). Thus, Model 2 had the pH recorded through the capsule in the stomach as dependent variable, as fixed factors the type of video clip and the number of the datapoints, and the by-subject intercepts as random effects. Models (1; 2; 5 and 6) are fully described in the Appendix 1. When statistically significant main or interaction effects were found, robust p-values were bootstrapped through the *mixed* function of the *afex* package (**Singmann et al., 2015b**). Post-hoc simple slope analyses of the statistically significant interactions were run using the *sim_slopes* function of the *interaction* package (**Long, 2019**). A third and fourth mixed-effect models were run to estimate the impact of EGG (normogastric and tachigastric respectively) peak frequency over the emotional VAS ratings. Those models featured *emotion* and *item* as fixed factors, but had the block-specific (centred and scaled) *EGG* (normogastric or tachigastric respectively) *peak frequencies* as continuous fixed factor and included by-subject intercepts as random effects.

For the detailed description of the models and the softwares/packages used, please see the Appendix 1.

## Acknowledgements

We thank Prof. Marco Tettamanti for sharing the full set of emotional stimuli, Danila Cosenza for her medical assistance, Luca Occhigrossi for technical support with the ingestibles, Maurizio Molisso for standardized meal preparation, our brave volunteers for their participation, and the members of AgliotiLab for providing helpful insights and advice on this study. Finally, we want to thanks the Reviewers (Dr. Ignacio Rebollo, Dr. Camilla North and the anonymous one) for providing very constructive feedback on this manuscript.

The support of the Institut d'études avancées de Paris to SMA is gratefully acknowledged. This research was supported by ERC Advanced Grant 789058 (eHonesty) to SMA; by Regione Lazio grant A0375-2020-36612 to SMA and MSP; and by Italian Ministry of Health Young Researcher Grant (2018-2367636 and 2021-12372815) to GP and MSP, and SEED PNR21 to GP.

## Additional information

### Funding

| Funder | Grant reference number | Author |
| --- | --- | --- |
| European Research Council | 789058 | Salvatore Maria Aglioti |
| Lazio Innova | A0375-2020-36612 | Maria Serena Panasiti Salvatore Maria Aglioti |
| Ministero della Salute | 2018-2367636 | Giuseppina Porciello |
| Ministero della Salute | 2021-12372815 | Giuseppina Porciello Maria Serena Panasiti |
| Sapienza Università di Roma | SEED PNR21 | Giuseppina Porciello |

The funders had no role in study design, data collection and interpretation, or the decision to submit the work for publication.

### Author contributions

Giuseppina Porciello, Conceptualization, Resources, Data curation, Formal analysis, Funding acquisition, Investigation, Visualization, Methodology, Writing - original draft, Project administration, Writing – review and editing; Alessandro Monti, Conceptualization, Data curation, Formal analysis, Investigation, Methodology, Project administration, Writing – review and editing; Maria Serena Panasiti, Conceptualization, Data curation, Formal analysis, Funding acquisition, Investigation, Methodology, Project administration, Writing – review and editing; Salvatore Maria Aglioti, Conceptualization, Supervision, Funding acquisition, Investigation, Methodology, Project administration, Writing – review and editing

### Author ORCIDs

Giuseppina Porciello ⓘ http://orcid.org/0000-0001-8374-5320
Salvatore Maria Aglioti ⓘ https://orcid.org/0000-0001-8175-7563

### Ethics

The protocol of this study was approved by the local Institutional Review Board (Fondazione Santa Lucia ethics committee, Protocol CE/PROG.636) and was performed in accordance with the Declaration of Helsinki (1991). All participants were naïve to the purpose of the experiment, provided their written informed consent before starting the experimental procedure, and received a reimbursement for their participation at the end of the study.

### Decision letter and Author response

Decision letter https://doi.org/10.7554/eLife.85567.sa1
Author response https://doi.org/10.7554/eLife.85567.sa2

## Additional files

### Supplementary files

• MDAR checklist

### Data availability

Data collected in this study are available at the OSF space dedicated to this project (https://osf.io/uq4t3/). Any additional information or material requests can be addressed to Giuseppina Porciello (giuseppina.porciello@uniroma1.it) or Salvatore Maria Aglioti (salvatoremaria.aglioti@uniroma1.it).

The following dataset was generated:

| Author(s) | Year | Dataset title | Dataset URL | Database and Identifier |
|---|---|---|---|---|
| Porciello G, Monti A, Panasiti MS, Aglioti SM | 2023 | Deep-body feelings: ingestible pills reveal gastric correlates of emotions | https://osf.io/uq4t3/ | Open Science Framework, 10.17605/OSF.IO/UQ4T3 |

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

# Appendix 1

## Supplementary Information about data analysis

### SmartPill data pre-processing

Once the pill was expelled and the receiver arrived in the lab, SmartPill data of each participant were uploaded and visualized through the MotiliGI software. Stomach, small bowel, large bowel, and whole gut transit times of the capsule (*Lee et al., 2014*) were calculated in order to estimate possible anomalies. All but one participant showed regular transit times. The remaining subject had an abnormal, that is too long large bowel transit time (>59 hr *Saad, 2016*). Consequently, his data were discarded from the statistical analyses. A custom MATLAB algorithm converted relative timestamps recorded by the SmartPill capsule in absolute times, so that each event (e.g. beginning and end of each experimental block of the emotional task) was paired to a definite hh:mm:ss:ms string.

### EGG data pre-processing

Raw EGG signals were visually inspected on LabChart (data analysis software, ADInstruments Ltd) to remove artifacts due to body movements.

For each emotional block, we calculated the individual EGG peak frequency, namely the maximum periodogram peak found in the 'normogastric' range, i.e. the range of frequencies that is compatible with the number of stomach contractions in healthy individuals (0.033–0.066 Hz ~2–4 cycles per minute, cpm *Wolpert et al., 2020*); as well as the one found in the 'tachigastric' range (0.067 Hz-0.167~4–10 cpm *Riezzo et al., 2013*).

EGG spectral density was computed using Welch's method on 200 s time windows with 150 s overlap (*Rebollo et al., 2018*). EGG analysis was performed with BrainVision Analyzer (Brain Products GmbH) and the MATLAB FieldTrip toolbox (*Oostenveld et al., 2011*).

### Mixed model analysis

Mixed models were specified as follows.

### Model 1 (SmartPill data, stomach)

*Emotional VAS* ratings ~*video clip content * item * (ph +pressure + temperature) + (1 | subject), data = stomach_data, control = lmerControl(optimizer = "nloptwrap", calc.deriv=FALSE).*

### Model 2 (SmartPill data, stomach)

Stomach pH ~video clip content * number of the datapoints ("Time") + (1|subject), data = stomach_data_all, control = lmerControl(optimizer = "nloptwrap", calc.deriv=FALSE).

### Model 3 (EGG data -normogastric band, stomach)

*Emotional VAS* ratings ~*video clip content * item * egg peak frequency_normogastric band + (1 | subject), data = egg_data, control = lmerControl(optimizer = "nloptwrap", calc.deriv=FALSE).*

### Model 4 (EGG data tachigastric band, stomach)

*Emotional VAS* ratings ~*video clip content * item * egg peak frequency_tachigastric band + (1 | subject), data = egg_data, control = lmerControl(optimizer = "nloptwrap", calc.deriv=FALSE).*

### Model 5 (SmartPill data, small bowel)

*Emotional VAS* ratings ~*video clip content * item * (ph +pressure + temperature) + (1 | subject), data = smallbowel_data, control = lmerControl(optimizer = "nloptwrap", calc.deriv=FALSE).*

### Model 6 (SmartPill data, large bowel)

*Emotional VAS* ratings ~*video clip content * item * (ph +pressure + temperature) + (1 | subject), data = largebowel_data, control = lmerControl(optimizer = "nloptwrap", calc.deriv=FALSE)*

## Statistical softwares and packages

Non parametric Friedman ANOVAs and parametric repeated measures ANOVAs followed by Bonferroni-corrected Wilcoxon matched-pairs tests were run using Statistica 7 software.

The remaining analyses were performed with R Studio. Specifically, we used the *lme4* package (*Bates et al., 2014*) to perform linear mixed-effects analyses; the *lmerTest* package (*Kuznetsova et al., 2017*) to extract p-values through a Type III analysis of variance with Satterthwaite's method; the *mixed* function of the *afex* package (*Singmann et al., 2015a*) to compute robust p-values with bootstrap method; the *interactions* (*Long, 2019*) and the *emmeans* (*Lenth et al., 2019*) packages to perform post-hoc simple slopes analysis and plots when significant interactions were found; and the *ggplots2* package (*Wickham, 2016*) to perform boxplots of the emotional, visceral and arousal experience. The standard assumptions and requirements of mixed models (linearity, homoscedasticity, absence of collinearity, and normality of residuals) were assessed through visual inspection of residual plots, the *shapiro.test* function and the *vif* function of the *car* package (*Fox and Weisberg, 2019*). The absence of singularity of the model was instead checked via the *check_singularity* function of the *performance* package (*Lüdecke et al., 2021*). Finally, the percentage of variance explained by each mixed-effects model was computed through the *r.squaredGLMM* function of Kamil Bartoń's *MuMIn* package.

## Supplementary results

### Subjective emotional experience

#### Capsule in the small bowel (Session 2)

The Friedman ANOVA comparing the perceived emotional experience triggered by the five types of video-clips in the second session (namely when the capsule was in the small bowel) was statistically significant ($\chi^2$(19)=358.763; p≤0.0001), suggesting that participants perceived different emotions after observing the different content of the video-clips. Planned post-hoc Bonferroni-corrected Wilcoxon matched-pairs tests showed results similar to those found in the first session. Specifically, VAS ratings of perceived disgust given after disgusting video-clips were higher than those given after all the other video-clips (all *Z*s ≥4.597; all ps ≤0.0001). VAS ratings of perceived fear given after fearful video-clips were higher than those given after all the other video-clips (all *Z*s ≥3.029; all ps ≤0.002). VAS ratings of perceived happiness given after happy video-clips were higher than those given after all the other video-clips (all *Z*s ≥4.372; all ps ≤0.0001). Finally, VAS ratings of perceived sadness given after sad video-clips were higher than those given after all the other video-clips (all *Z*s ≥4.509; all ps ≤0.0001). See *Appendix 1—figure 1* for a graphical illustration of these results.

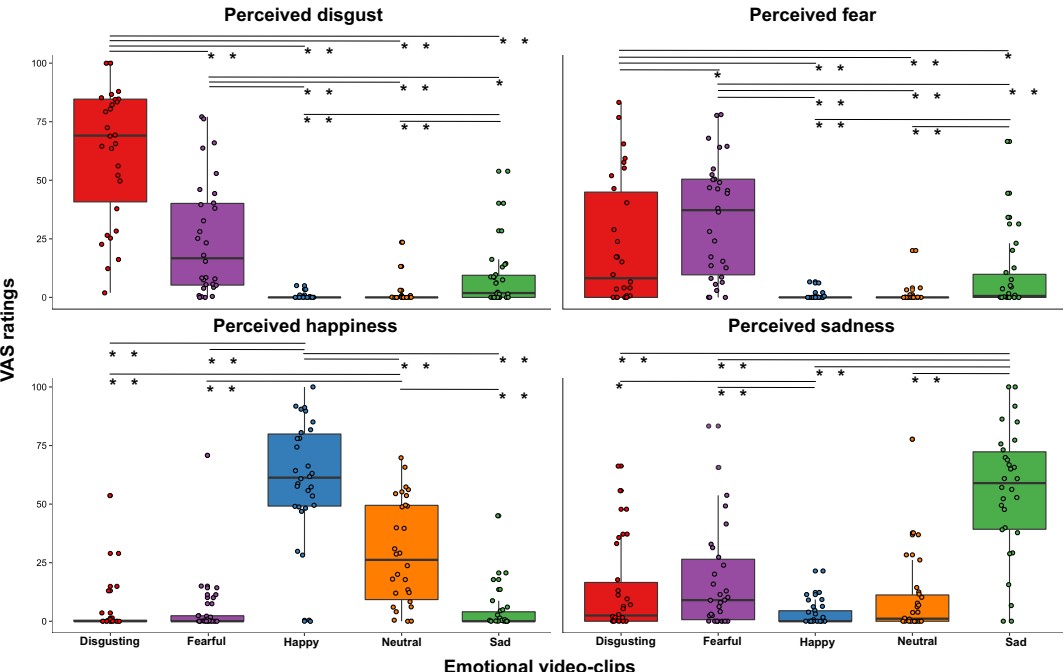

**Appendix 1—figure 1.** Perceived emotions results (pill in the small bowel, session 2). Perceived emotions (disgust, fear, happiness, and sadness) measured using 0–100 visuo-analogue scale (VAS) ratings, as a function of the five categories of video-clips (disgusting, fearful, happy, sad, and neutral) shown during the second session of this study (i.e. when the capsule was in the small bowel). Boxplots are drawn from the first to the third quartiles, with the horizontal lines representing the medians. The lines extending from the top and bottom of the boxplots represent the smallest and largest values within 1.5 times the interquartile ranges (IQR) from the first and third quartiles, respectively. * p≤0.05; ** p≤0.01.

## Capsule in the large bowel (Session 3)

The Friedman ANOVA comparing the perceived emotional experience triggered by the five types of video-clips when the capsule was in the large bowel was statistically significant ($\chi^2(19)=328.053$; p≤0.0001), suggesting that participants perceived different emotions even after observing for three times the different content of the same video-clips. Planned post-hoc Bonferroni-corrected Wilcoxon matched-pairs tests showed results akin to those found in the first and second session. Specifically, VAS ratings of perceived disgust given after disgusting video-clips were higher than those given after all the other video-clips (all $Zs \geq 4.421$; all ps ≤0.0001). VAS ratings of perceived fear given after fearful video-clips were higher than those given after all the other video-clips (all $Zs \geq 3.484$; all ps ≤0.0005). VAS ratings of perceived happiness given after happy video-clips were higher than those given after all the other video-clips (all $Zs \geq 4.379$; all ps ≤0.0001). Finally, VAS ratings of perceived sadness given after sad video-clips were higher than those given after all the other video-clips (all $Zs \geq 4.660$; all ps ≤0.0001). See *Appendix 1—figure 2* for a graphical illustration of these results.

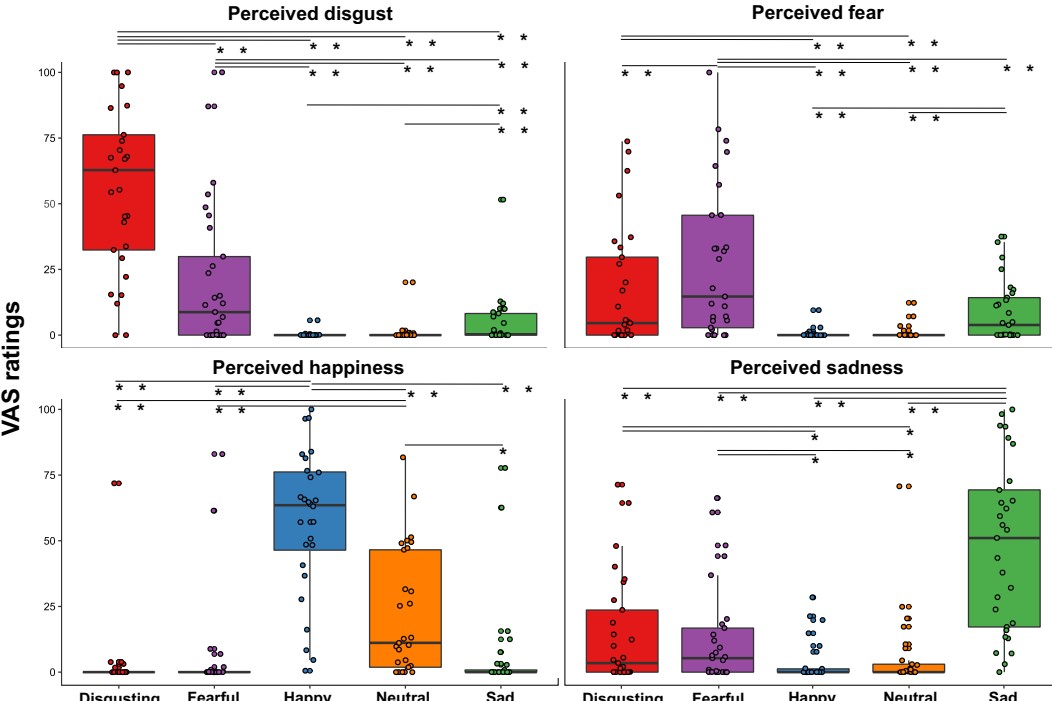

**Appendix 1—figure 2.** Perceived emotions results (pill in the large bowel, session 3). Perceived emotions (disgust, fear, happiness, and sadness) measured using 0–100 visuo-analogue scale (VAS) ratings, as a function of the five categories of video-clips (disgusting, fearful, happy, sad, and neutral) shown during the third session of this study (i.e. when the capsule was in the large bowel). Boxplots are drawn from the first to the third quartiles, with the horizontal lines representing the medians. The lines extending from the top and bottom of the boxplots represent the smallest and largest values within 1.5 times the interquartile ranges (IQR) from the first and third quartiles, respectively. * p≤0.05; ** p≤0.01.

## Subjective visceral experience

### Capsule in the small bowel (Session 2)

The Friedman ANOVA comparing the perceived visceral experience (i.e. gastric, cardiac, respiratory sensations and arousal) triggered by the five types of video-clips in the second session (namely when the capsule was in the small bowel) was statistically significant ($\chi^2$(19)=289.385; p≤0.0001), suggesting that participants perceived different sensations after observing the different content of the video-clips. Planned post-hoc Bonferroni-corrected Wilcoxon matched-pairs tests showed results akin to those found in the first session. Specifically, *gastric sensations* were higher when participants were asked to observe disgusting and fearful videos. In particular, gastric sensations evoked by disgusting videos were significantly higher than those evoked by all the other video-clips (all Zs ≥2.911; all ps ≤0.005), while gastric sensations evoked by fearful video-clips were higher than those evoked by happy and neutral video-clips (all Zs ≥2.58; all ps ≤0.01). As to the *cardiac sensations*, we found that all the emotional video-clips evoked higher cardiac sensations with respect to the neutral video-clips (all Zs ≥2.501; all ps ≤0.012) even though happy video-clips were only marginally significant considering the Bonferroni corrected significance threshold of 0.0125 (Z=2.352.; P=0.019). As for the *respiratory sensations*, we found that all the emotional video-clips but happy video-clips evoked higher respiratory sensations with respect to the neutral video-clips (all Zs ≥3.027; all ps ≤0.002). Moreover, respiratory sensations evoked by disgusting and sad videos were also significantly higher than those evoked by happy video-clips (all Zs ≥2.613; all ps ≤0.009). Finally, we found that all the emotional video-clips evoked higher *feeling of arousal* with respect to the neutral video-clips, (all Zs ≥4.28; all ps ≤0.0001). See ***Appendix 1—figure 3*** for a graphical illustration of these results.

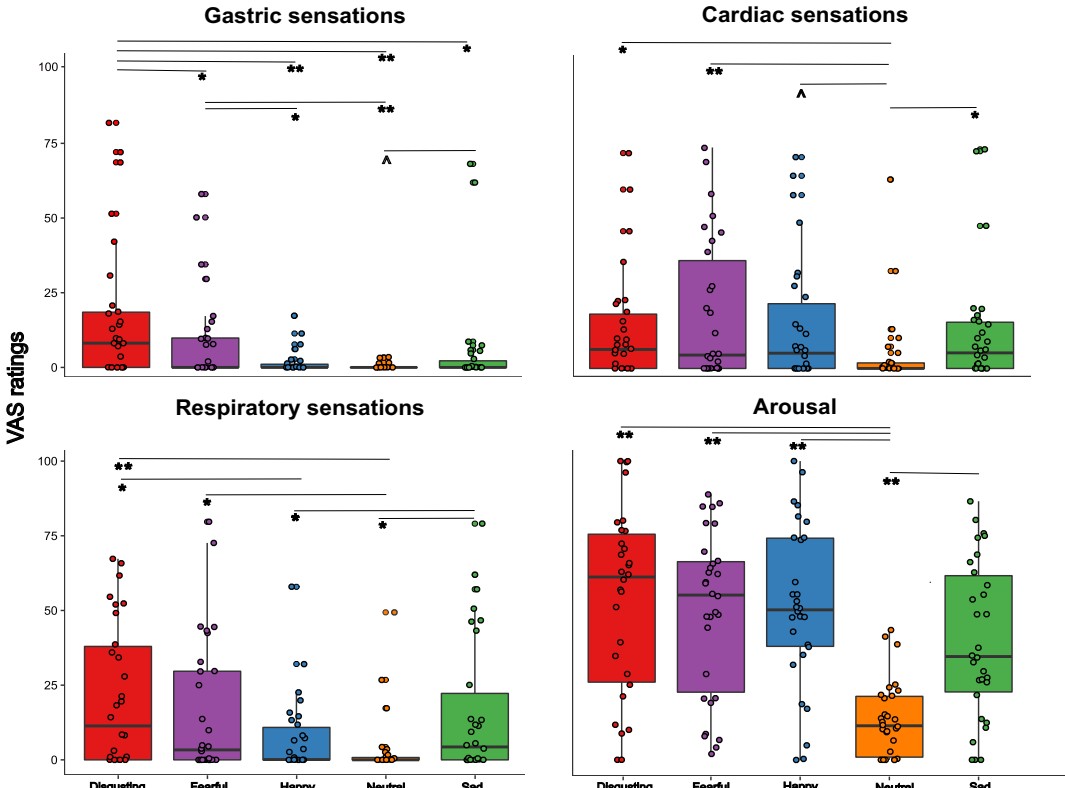

**Appendix 1—figure 3.** Visceral sensations results (pill in the small bowel, session 2). Perceived visceral sensations (gastric, cardiac, and respiratory) and arousal, measured using 0–100 visuo-analogue scale (VAS) ratings, as a function of the five categories of video-clips (disgusting, fearful, happy, sad, and neutral) shown during the second session of this study (i.e. when the capsule was in the small bowel). Boxplots are drawn from the first to the third quartiles, with the horizontal lines representing the medians. The lines extending from the top and bottom of the boxplots represent the smallest and largest values within 1.5 times the interquartile ranges (IQR) from the first and third quartiles, respectively. ^ p=0.019; * p≤0.01; ** p≤0.001.

## Capsule in the large bowel (Session 3)

The Friedman ANOVA comparing the perceived visceral experience triggered by the five types of video-clips when the capsule was in the large bowel was statistically significant ($\chi^2$(19)=286.999; p≤0.0001), suggesting that participants perceived different sensations after observing the different content of the video-clips. *Gastric sensations* were higher when participants were asked to observe disgusting and fearful videos. In particular, gastric sensations evoked by disgusting videos were significantly higher than those evoked by all the other video-clips (all *Z*s ≥2.656; all ps ≤0.008), while gastric sensations evoked by fearful video-clips were higher than those evoked by neutral video-clips (*Z*≥2.667; p≤0.008). As to the *cardiac sensations*, we found that all the emotional video-clips evoked higher cardiac sensations with respect to the neutral video-clips (all Zs ≥3.419; all ps ≤0.001). As for the *respiratory sensations*, we found that all the emotional video-clips evoked higher respiratory sensations with respect to the neutral video-clips (all Zs ≥2.844; all ps ≤0.004). Moreover, respiratory sensations evoked by disgusting and sad videos were also significantly higher than those evoked by happy video-clips (all Zs ≥2.632; all ps ≤0.008). Finally, we found that all the emotional video-clips evoked higher *feeling of arousal* with respect to the neural video-clips, (all Zs ≥3.575; all ps ≤0.001). See *Appendix 1—figure 4* for a graphical illustration of these results.

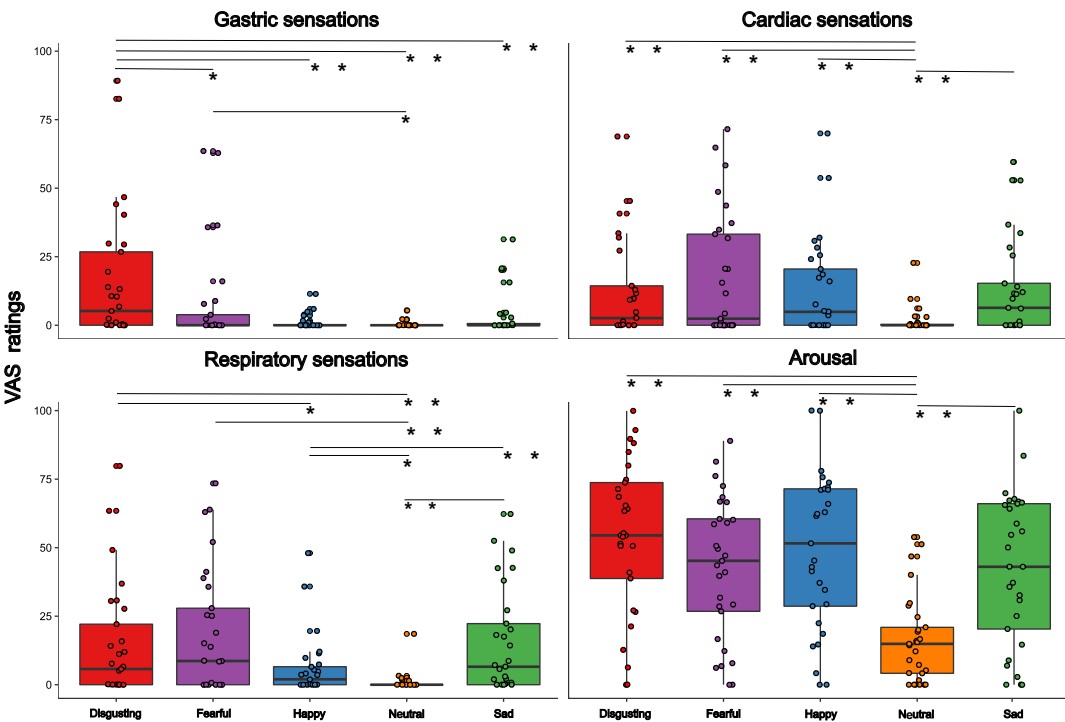

**Appendix 1—figure 4.** Visceral sensations results (pill in the large bowel, session 3). Perceived visceral sensations (gastric, cardiac and respiratory) and arousal, measured using 0–100 visuo-analogue scale (VAS) ratings, as a function of the five categories of video-clips: disgusting, fearful, happy, sad, and neutral shown during the third session of this study (i.e. when the capsule was in the large bowel). Boxplots are drawn from the first to the third quartiles, with the horizontal lines representing the medians. The lines extending from the top and bottom of the boxplots represent the smallest and largest values within 1.5 times the interquartile ranges (IQR) from the first and third quartiles, respectively. ^ p=0.018; * p≤0.05; ** p≤0.01.

## Gut markers of perceived emotions: pH, temperature, pressure and EGG peak frequency

### Capsule in the stomach (Session 1): pill-related gastric markers (pH, temperature, pressure)

Type III analysis of variance of Model 1 showed a statistically significant 2-way interaction between item and gastric pH, suggesting that emotional experience reported by participants on the VAS ratings varied according to the pH of the stomach and the type of perceived emotion, irrespectively of the type of observed video clip. The follow-up post hoc simple slope analysis showed that the lower (i.e. more acidic) was the pH of the stomach, the more our participants reported feeling of disgust and fear, while the higher was the pH of their stomach (i.e. less acidic) the more they reported feelings of happiness, see *Appendix 1—figure 5* for a graphical representation of these effects. For the detailed description of Model 1 results refer to *Appendix 1—table 1*, to see results of the more conservative version of Model 1 see *Appendix 1—table 2* below.

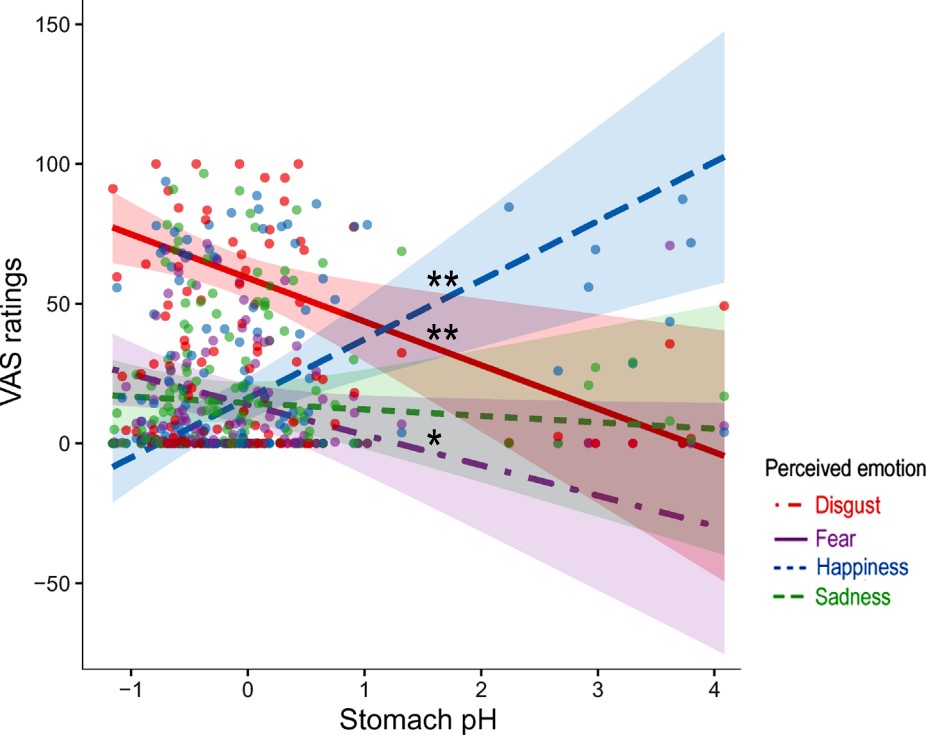

**Appendix 1—figure 5.** Stomach pH influences perceived emotions (two-way interaction between item and gastric pH). Effects of stomach pH on perceived emotions (disgust, fear, happiness, and sadness). * p≤0.05; ** p≤0.01.

## Capsule in the stomach (Session 1): EGG-related gastric markers (normogastric and tachigastric EGG peak frequencies)

The repeated measures ANOVA performed on the individual peak frequencies with the different content of the video clips as the only within subjects factor run on the tachigastric cycle did show a main effect of the type of video clip ($F_{(4, 112)}=2.907$, $p=0.025$, Eta2 (partial)=0.09). However, the Bonferroni corrected post hoc tests did not show any significant difference between the different type of emotional video clips and the neutral condition. The only significant comparison was the one between happy and fearful video clips, namely participants' tachigastric cycle was faster when they observed happy vs fearful video clips ($p=0.038$) see *Appendix 1—figure 6* below for a graphical representation of the results.

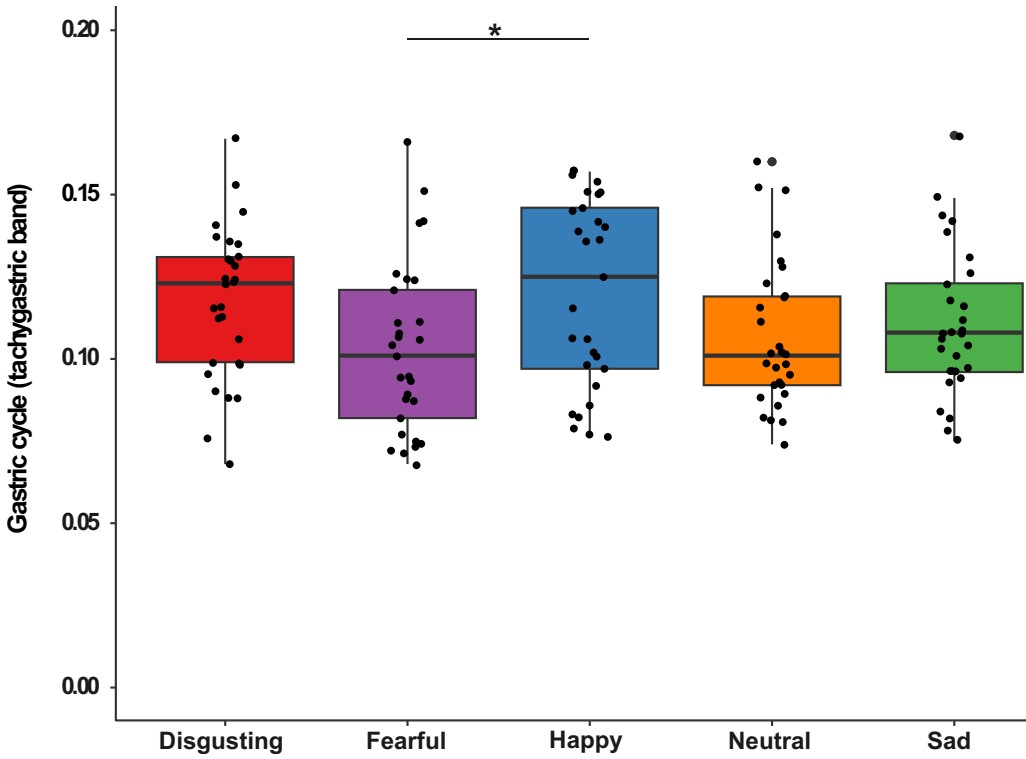

**Appendix 1—figure 6.** Gastric cycle (tachigastric band) results. Gastric cycle calculated in the tachigastric band as a function of the five categories of video-clips: disgusting, fearful, happy, sad, and neutral shown during the first session of this study (i.e., when the capsule was in the stomach). Boxplots are drawn from the first to the third quartiles, with the horizontal lines representing the medians. The lines extending from the top and bottom of the boxplots represent the smallest and largest values within 1.5 times the interquartile ranges (IQR) from the first and third quartiles, respectively. * p≤0.05.

## Capsule in the small bowel (Session 2)

Model 5 (see the main text for data analysis procedure) did not show any problem of singular fit, had a marginal $R^2$=0.57 and a conditional $R^2$=0.66. Visual inspection of the plots did not reveal any obvious deviation from homoscedasticity. Residuals were not normally distributed according to Shapiro-Wilk normality test, but linear models are robust against violations of normality (*Gelman and Hill, 2006*). As for collinearity, all independent variables had a (GVIF^(1/ (2*Df)))^2<10.

Type III analysis of variance of Model 5 showed a statistically significant two-way interaction between video-clip content and item (*F*=65.29, p<0.005), but no statistically significant three-way interactions between video-clip content, item, and small bowel pH (*F*=0.60, p=0.84), or video-clip content, item, and small bowel pressure (*F*=1.44, p=0.14), or video-clip content, item, and small bowel temperature (*F*=1.65, p=0.07). This pattern of results suggests that emotional experience reported by participants on the VAS was only influenced by the content of the projected video-clips and the item type, but not by pH, pressure, or temperature recorded in the small bowel. For a detailed report of the model results please refer to the *Appendix 1—table 6*

## Capsule in the large bowel (Session 3)

Model 6 did not show any problem of singular fit, had a marginal $R^2$=0.45 and a conditional $R^2$=0.58. Visual inspection of the plots did not reveal any obvious deviation from homoscedasticity. Residuals were not normally distributed according to Shapiro-Wilk normality test, but linear models are robust against violations of normality (*Gelman and Hill, 2006*). As for collinearity, all independent variables had a (GVIF^(1/ (2*Df)))^2<10.

Type III analysis of variance of Model 6 showed a statistically significant two-way interaction between video-clip content and item (*F*=42.46, p<0.005), but any statistically significant three-way

interactions between video-clip content, item and large bowel pH (*F*=0.61, p=0.83), or video-clip content, item and large bowel pressure (*F*=0.4, p=0.96), or video-clip content, item and large bowel temperature (*F*=0.56, p=0.88), suggesting that emotional experience reported by participants on the VAS was only influenced by the content of the projected video-clips and the item type, but not by pH, pressure, or temperature recorded in the large bowel. For a detailed report of the model results please refer to *Appendix 1—table 7*.

**Appendix 1—table 1.** Results of Model 1 (pill in the stomach, session 1).
Perceived emotions predicted by the video clip content, type of item and stomach pH, temperature and pressure. Type III analysis of variance table with Satterthwaite's method.

| | Sum Sq | Mean Sq | NumDF | DenDF | F value | p |
|---|---|---|---|---|---|---|
| Video-clip content | 14864 | 3715.9 | 4 | 460.33 | 11.8690 | 3.475e-09*** |
| Item | 12406 | 4135.2 | 3 | 453.84 | 13.2081 | 2.785e-08*** |
| (Stomach) Pressure | 1321 | 1320.6 | 1 | 286.71 | 4.2182 | 0.04090* |
| (Stomach) pH | 80 | 79.7 | 1 | 96.07 | 0.2545 | 0.61511 |
| (Stomach) T | 314 | 314.3 | 1 | 31.69 | 1.0039 | 0.32397 |
| Video-clip content:Item | 154608 | 12884.0 | 12 | 453.84 | 41.1526 | <2.2e-16*** |
| Video-clip content:(Stomach) Pressure | 918 | 229.4 | 4 | 477.11 | 0.7327 | 0.56993 |
| Video-clip content:(Stomach) pH | 2335 | 583.8 | 4 | 471.88 | 1.8646 | 0.11549 |
| Video-clip content:(Stomach) T | 1905 | 476.3 | 4 | 459.68 | 1.5215 | 0.19478 |
| Item:(Stomach) Pressure | 1262 | 420.6 | 3 | 453.84 | 1.3433 | 0.25969 |
| Item:(Stomach) pH | 7715 | 2571.7 | 3 | 453.84 | 8.2142 | 2.470e-05*** |
| Item:(Stomach) T | 1216 | 405.3 | 3 | 453.84 | 1.2946 | 0.27568 |
| Video-clip content:item:(Stomach) Pressure | 4093 | 341.1 | 12 | 453.84 | 1.0895 | 0.36686 |
| Video-clip content:item:(Stomach) pH | 7433 | 619.4 | 12 | 453.84 | 1.9784 | 0.02456* |
| Video-clip content:item:(Stomach) T | 2293 | 191.0 | 12 | 453.84 | 0.6102 | 0.83407 |

**Appendix 1—table 2.** Results of the Bonferroni corrected post hoc tests following the triple interaction Video-clip content*Item*Stomach pH of Model 1.
It has been performed, via *emtrends* function, following the significant triple interaction Video-clip content:item:(Stomach) pH, obtained by running the more conservative version of Model 1.

| Video-clip content | Item | pH | SE | df | T ratio | p |
|---|---|---|---|---|---|---|
| Disgust | Disgust | −15.64 | 5.75 | 159.00 | −2.72 | 0.01 ** |
| Fearful | Disgust | 6.02 | 3.35 | 152.00 | 1.80 | 0.07 |
| Happy | Disgust | −2.18 | 2.87 | 235.00 | −0.76 | 0.45 |
| Neutral | Disgust | −1.68 | 5.17 | 209.00 | −0.32 | 0.75 |
| Sad | Disgust | 2.98 | 4.38 | 191.00 | 0.68 | 0.50 |
| Disgust | Happy | 21.12 | 5.75 | 159.00 | 3.67 | <0.001*** |
| Fearful | Happy | 7.56 | 3.35 | 152.00 | 2.26 | 0.03* |
| Happy | Happy | 5.94 | 2.87 | 235.00 | 2.08 | 0.04* |
| Neutral | Happy | 3.55 | 5.17 | 209.00 | 0.69 | 0.49 |
| Sad | Happy | 6.04 | 4.38 | 191.00 | 1.38 | 0.17 |
| Disgust | Sad | −2.34 | 5.75 | 159.00 | −0.41 | 0.68 |
| Fearful | Sad | 1.81 | 3.35 | 152.00 | 0.54 | 0.59 |

*Appendix 1—table 2 Continued on next page*

*Appendix 1—table 2 Continued*

| Video-clip content | Item | pH | SE | df | T ratio | p |
|---|---|---|---|---|---|---|
| Happy | Sad | 0.02 | 2.87 | 235.00 | 0.01 | 0.99 |
| Neutral | Sad | –6.33 | 5.17 | 209.00 | –1.22 | 0.22 |
| Sad | Sad | 4.68 | 4.38 | 191.00 | 1.07 | 0.29 |
| Disgust | Fear | –10.90 | 5.75 | 159.00 | –1.90 | 0.06 |
| Fearful | Fear | 1.02 | 3.35 | 152.00 | 0.31 | 0.76 |
| Happy | Fear | –0.40 | 2.87 | 235.00 | –0.14 | 0.89 |
| Neutral | Fear | –0.98 | 5.17 | 209.00 | –0.19 | 0.85 |
| Sad | Fear | 1.14 | 4.38 | 191.00 | 0.26 | 0.80 |

**Appendix 1—table 3.** Results of Model 2 (pill in the stomach, session 1).
Stomach pH predicted by the video clip content and the time (i.e. number of datapoint). In this case all the datapoints collected via the Smartpill have been used. Type III analysis of variance table with Satterthwaite's method.

| | Sum Sq | Mean Sq | NumDF | DenDF | F value | p |
|---|---|---|---|---|---|---|
| Video-clip content | 25.5256 | 6.3814 | 4 | 6043.1 | 22.242 | <2.2e-16*** |
| Time (datapoint) | 5.8059 | 5.8059 | 1 | 6043.2 | 20.237 | 6.971e-06*** |
| Video-clip content:Time (datapoint) | 3.7297 | 0.9324 | 4 | 6043.1 | 3.250 | 0.01133* |

**Appendix 1—table 4.** Results of the Bonferroni corrected post hoc tests following the double interaction Video-clip content*Item of Model 2.

| Video-clip content | trend | SE | df | T ratio | p |
|---|---|---|---|---|---|
| Disgust | –0.001539 | 0.00121 | 6043 | –1.269 | 0.2045 |
| Fearful | –0.005208 | 0.00118 | 6043 | –4.421 | <0.0001 |
| Happy | 0.000528 | 0.00118 | 6043 | 0.447 | 0.6552 |
| Neutral | –0.003402 | 0.00122 | 6043 | –2.796 | 0.0052 |
| Sad | –0.002463 | 0.00120 | 6043 | –2.049 | 0.0405 |

**Appendix 1—table 5.** Results of Model 3 (EGG peak frequency_normogastric band, session 1).
Perceived emotions predicted by the video clip content, type of item and EGG normogastric peak frequency. Type III analysis of variance table with Satterthwaite's method.

| | Sum Sq | Mean Sq | NumDF | DenDF | F value | p |
|---|---|---|---|---|---|---|
| Video-clip content | 13617 | 3404.3 | 4 | 418.86 | 10.1660 | 7.302e-08*** |
| Item | 8180 | 2726.6 | 3 | 405.51 | 8.1422 | 2.819e-05*** |
| Egg peak frequency | 466 | 465.9 | 1 | 402.48 | 1.3913 | 0.2389 |
| Video-clip content:item | 142599 | 11883.3 | 12 | 405.51 | 35.4858 | <2.2e-16*** |
| Video-clip content:egg peak frequency | 823 | 205.7 | 4 | 425.15 | 0.6142 | 0.6526 |
| Item:egg peak frequency | 522 | 173.8 | 3 | 405.51 | 0.5191 | 0.6693 |
| Video-clip content:item:egg peak frequency | 2282 | 190.2 | 12 | 405.51 | 0.5679 | 0.8680 |

**Appendix 1—table 6.** Results of Model 5 (pill in the small bowel, session 2).
Perceived emotions predicted by the video clip content, type of item and small bowel pH, temperature and pressure. Type III analysis of variance table with Satterthwaite's method.

| | Sum Sq | Mean Sq | NumDF | DenDF | F value | P |
|---|---|---|---|---|---|---|
| Video-clip content | 13778 | 3444.6 | 4 | 478.81 | 12.9585 | 5.013e-10*** |
| Item | 3314 | 1104.6 | 3 | 469.26 | 4.1554 | 0.006374** |
| (Small bowel) Pressure | 78 | 77.7 | 1 | 140.02 | 0.2923 | 0.589630 |
| (Small bowel) pH | 805 | 805.1 | 1 | 69.10 | 3.0289 | 0.086244 |
| (Small bowel) T | 43 | 42.7 | 1 | 30.06 | 0.1607 | 0.691327 |
| Video-clip content:Item | 208270 | 17355.8 | 12 | 469.26 | 65.2925 | <2.2e-16*** |
| Video-clip content:(Small bowel) Pressure | 1003 | 250.7 | 4 | 486.53 | 0.9431 | 0.438650 |
| Video-clip content:(Small bowel) pH | 778 | 194.4 | 4 | 477.46 | 0.7314 | 0.570808 |
| Video-clip content:(Small bowel) T | 181 | 45.4 | 4 | 471.94 | 0.1707 | 0.953324 |
| Item:(Small bowel) Pressure | 1160 | 386.7 | 3 | 469.26 | 1.4546 | 0.226235 |
| Item:(Small bowel) pH | 610 | 203.5 | 3 | 469.26 | 0.7655 | 0.513789 |
| Item:(Small bowel) T | 196 | 65.2 | 3 | 469.26 | 0.2452 | 0.864744 |
| Video-clip content:item:(Small bowel) Pressure | 4598 | 383.2 | 12 | 469.26 | 1.4416 | 0.143390 |
| Video-clip content:item:(Small bowel) pH | 1930 | 160.8 | 12 | 469.26 | 0.6050 | 0.838509 |
| Video-clip content:item:(Small bowel) T | 5273 | 439.4 | 12 | 469.26 | 1.6530 | 0.074340 |

**Appendix 1—table 7.** Results of Model 6 (pill in the large bowel, session 3).
Perceived emotions predicted by the video clip content, type of item and large bowel pH, temperature and pressure. Type III analysis of variance table with Satterthwaite's method.

| | Sum Sq | Mean Sq | NumDF | DenDF | F value | P |
|---|---|---|---|---|---|---|
| Video-clip content | 14179 | 3544.7 | 4 | 473.75 | 11.3579 | 8.251e-09 *** |
| Item | 4319 | 1439.7 | 3 | 468.86 | 4.6129 | 0.003418 ** |
| (Large bowel) Pressure | 3 | 2.8 | 1 | 269.21 | 0.0089 | 0.924857 |
| (Large bowel) pH | 0 | 0.0 | 1 | 58.50 | 0.0000 | 0.994501 |
| (Large bowel) T | 33 | 33.1 | 1 | 31.27 | 0.1061 | 0.746856 |
| Video-clip content:Item | 159021 | 13251.8 | 12 | 468.86 | 42.4611 | < 2.2e-16 *** |
| Video-clip content:(Large bowel) Pressure | 1165 | 291.2 | 4 | 485.30 | 0.9331 | 0.444414 |
| Video-clip content:(Large bowel) pH | 973 | 243.4 | 4 | 474.95 | 0.7798 | 0.538680 |
| Video-clip content:(Large bowel) T | 720 | 180.1 | 4 | 472.93 | 0.5770 | 0.679421 |
| Item:(Large bowel) Pressure | 343 | 114.2 | 3 | 468.86 | 0.3659 | 0.777669 |
| Item:(Large bowel) pH | 755 | 251.8 | 3 | 468.84 | 0.8069 | 0.490491 |
| Item:(Large bowel) T | 938 | 312.6 | 3 | 468.86 | 1.0017 | 0.391836 |
| Video-clip content:item:(Large bowel) Pressure | 1496 | 124.7 | 12 | 468.86 | 0.3994 | 0.963735 |
| Video-clip content:item:(Large bowel) pH | 2288 | 190.7 | 12 | 468.84 | 0.6109 | 0.833531 |
| Video-clip content:item:(Large bowel) T | 2089 | 174.1 | 12 | 468.86 | 0.5578 | 0.875755 |

**Appendix 1—table 8.** Emotional video-clips content.
List of emotional video-clips used in the emotional induction task with a brief description in the second column, adapted by *Tettamanti et al., 2012*.

| Category_Number of the Video-clips | Description of the Video-clips |
|---|---|
| Disgust_1 | A monstrous creature vomits a corrosive fluid on a dead man's face and dissolves it |

*Appendix 1—table 8 Continued on next page*

*Appendix 1—table 8 Continued*

| Category_Number of the Video-clips | Description of the Video-clips |
|---|---|
| Disgust_2 | A severed human head is covered with insects |
| Disgust_3 | A man is covered with faecal matter |
| Disgust_4 | Some dishes and a wall are covered with insects |
| Disgust_5 | A man takes a worm off an animal's wound and holds it |
| Disgust_6 | A monstrous creature vomits a corrosive fluid on a dead man's foot |
| Disgust_7 | A monstrous creature vomits a corrosive fluid on a dead man's hand and dissolves it |
| Disgust_8 | A man plunges his hand in a toilet, retracts it covered with faecal matter and almost vomits |
| Disgust_9 | A man removes part of his own skin |
| Disgust_10 | A man vomits in a bowl, a man watches it and another man smells it |
| Disgust_11 | A man takes off his fingernail |
| Disgust_12 | A man eats snails |
| Disgust_13 | Close up on a fly which approaches a severed human hand covered with insects |
| Disgust_14 | Close up on insects |
| Disgust_15 | A man smells a bowl of vomit and drinks it |
| Disgust_16 | A girl vomits on a man's face |
| Disgust_17 | A man squeezes a purulent dot on his hand |
| Disgust_18 | A man is covered with a substance and tries to remove it from his skin |
| Disgust_19 | Some surgeons perform a surgical operation on a bloodied inhuman creature |
| Disgust_20 | A monstrously shaped egg cracks open and viscous substances spill out |
| Disgust_21 | A bloodied girl whose upper lip is missing bites a man's neck |
| Disgust_22 | A purulent inhuman organ is being dissected |
| Disgust_23 | A woman vomits on a man's face |
| Disgust_24 | A monstrous creature is dissected |
| Fear_1 | A girl is approached by a scary creature |
| Fear_2 | A bear roars against a man |
| Fear_3 | A man attacks a woman |
| Fear_4 | A woman is approached by a monster |
| Fear_5 | Several men are falling from a mountain |
| Fear_6 | A woman is held underwater by a scary creature |
| Fear_7 | A woman tries to escape and she is stabbed in her hand |
| Fear_8 | A woman is attacked by a man |
| Fear_9 | A woman is attacked by a scary creature |
| Fear_10 | A man is attacked by a shark |
| Fear_11 | An aeroplane is disrupted during flight |

*Appendix 1—table 8 Continued on next page*

*Appendix 1—table 8 Continued*

| Category_Number of the Video-clips | Description of the Video-clips |
| --- | --- |
| Fear_12 | A woman is attacked by an alien |
| Fear_13 | A man is attacked by a shark |
| Fear_14 | A man is tied to a chair with his mouth duct taped |
| Fear_15 | Two climbers hang in the void |
| Fear_16 | A woman is attacked by a man |
| Fear_17 | A woman falls from a mountain |
| Fear_18 | A woman falls from a building |
| Fear_19 | A man is attacked by a man |
| Fear_20 | A man at risk of being crashed |
| Fear_21 | Two man hang on a building ledge |
| Fear_22 | A woman is approached by a monster |
| Fear_23 | A woman is attacked by a man |
| Fear_24 | A man is trapped underwater |
| Happy_1 | Several persons rejoice at a soccer match |
| Happy_2 | Several boys and a man rejoice after scoring a goal in a soccer match |
| Happy_3 | A woman smiles after seeing an old lady beating a young man with her purse |
| Happy_4 | A child smiles while looking at two persons smiling and hugging each other |
| Happy_5 | A woman smiles while looking at a man playing with a child |
| Happy_6 | A woman and a man kiss each other, women claps their hands |
| Happy_7 | A woman and a man run on a shore playing with each other |
| Happy_8 | A man rejoices after scoring a goal in a soccer match |
| Happy_9 | A woman and a man kiss each other |
| Happy_10 | A woman and a man kiss each other |
| Happy_11 | A woman and a man play together |
| Happy_12 | A man and a woman laugh together |
| Happy_13 | A man hugs a woman |
| Happy_14 | A man and a woman smile while looking at a dog lactating her puppies |
| Happy_15 | Several persons dance and a girl laughs |
| Happy_16 | A woman and a man laugh together |
| Happy_17 | A girl opens a present, a woman kisses her |
| Happy_18 | A man scores a goal in a soccer match |
| Happy_19 | A man hugs his sons |
| Happy_20 | A woman and a man dance and smile at each other |
| Happy_21 | Several persons rejoice at a soccer match |
| Happy_22 | Players celebrating for scoring a goal |
| Happy_23 | Several persons rejoice at a soccer match |

*Appendix 1—table 8 Continued on next page*

*Appendix 1—table 8 Continued*

| Category_Number of the Video-clips | Description of the Video-clips |
|---|---|
| Happy_24 | Several persons rejoice at a rugby match |
| Sad_1 | A woman cries and holds the hand of a man in a hospital bed |
| Sad_2 | A man cries while holding a dead woman |
| Sad_3 | Two persons are separated as the train leaves |
| Sad_4 | A sick child is in a hospital bed |
| Sad_5 | A man cries while holding a dead woman |
| Sad_6 | A woman kisses the forehead of a dead boy |
| Sad_7 | A woman cries on the phone |
| Sad_8 | A woman cries while holding a dead man |
| Sad_9 | Dead bodies are moved by the waves on the shore |
| Sad_10 | A man cries while holding the bloodied body of a man |
| Sad_11 | A woman cries and hugs a man |
| Sad_12 | A sick woman is in bed, a man holds her hand |
| Sad_13 | A man crying while looking at a dead man |
| Sad_14 | A woman cries, two women hug her |
| Sad_15 | A man holds a dead man |
| Sad_16 | Several dead bodies lay upon the ground |
| Sad_17 | Several persons are in a hospital bed |
| Sad_18 | Dead bodies lay on a shore |
| Sad_19 | Several persons watch three coffins on the ground |
| Sad_20 | A woman and a man hold each other |
| Sad_21 | Three persons cry while holding each other |
| Sad_22 | A man looks upon a dead body |
| Sad_23 | A woman closes herself in a room and cries |
| Sad_24 | A woman lays a flower on a coffin at a funeral, another woman watches |
| Neutral_1 | Two men, one speaking and the other one listening |
| Neutral_2 | A woman opening the window of the dog kennel in an old mansion |
| Neutral_3 | A woman meets a man at the train station |
| Neutral_4 | A woman takes the computer from a bag and puts it on a table |
| Neutral_5 | Several persons observe drawings and sculptures |
| Neutral_6 | Several persons talk and walk in an office |
| Neutral_7 | A woman walks in a large building, several other persons in the background |
| Neutral_8 | Two waiters greet several persons entering a room |
| Neutral_9 | A man closes the door and walks away |
| Neutral_10 | A man works in a building site, several other men working in the background |
| Neutral_11 | A woman in a boat sits and opens a book |

*Appendix 1—table 8 Continued*

| Category_Number of the Video-clips | Description of the Video-clips |
|---|---|
| Neutral_12 | Several persons walking in a corridor, a woman enters an office |
| Neutral_13 | Men carving and moving large cinematographic props |
| Neutral_14 | A woman and a man speak while cleaning the kitchen |
| Neutral_15 | A woman walks down the road, some children on bikes wave at her |
| Neutral_16 | A women's meeting, one speaks while the others nod |
| Neutral_17 | A woman hoovers a room |
| Neutral_18 | A man walks in the countryside while reading |
| Neutral_19 | A man writes while sitting on the outside of a ferry |
| Neutral_20 | Different scenes from a food market |
| Neutral_21 | A woman types at the computer and lifts the receiver |
| Neutral_22 | A man and two young men talk by the phone |
| Neutral_23 | A man drawing |
| Neutral_24 | A man cleans the leaves off the yard |

