## [Editor Report]

This important study used a novel method to relate gastric acidity to subjective ratings of emotions induced by video clips. The findings are convincing, have broad implications for the field of emotion research, and open new avenues of research for understanding psychosomatic disorders.

---

## [Decision Letter]

**Decision letter after peer review:**

Thank you for submitting your article "Deep-body feelings: ingestible pills reveal gastric correlates of emotions" for consideration by *eLife*. Your article has been reviewed by 3 peer reviewers, one of whom is a member of our Board of Reviewing Editors, and the evaluation has been overseen by Christian Büchel as the Senior Editor. The following individuals involved in review of your submission have agreed to reveal their identity: Ignacio Rebollo (Reviewer #1); Camilla L. Nord (Reviewer #2).

Essential revisions (for the authors):

1) Include more information about the methods and statistical analyses.

2) Provide more support for the main findings (e.g., ruling out influence of outliers, false positives, or specific analysis choices), including changes to Figure 4.

3) Include effect sizes

4) Conduct additional analyses to gain more insight into possible mechanisms, e.g., on the EGG data or the relation between subjective and objective measures

5) The reviewers also make useful suggestions for streamlining the presentation of the results.

*Reviewer #1 (Recommendations for the authors):*

Below suggestions that I believe could improve the impact of the paper.

The authors explore the link between multiple emotions, subjective reports, gut compartments, and multiple gut indices, but the main finding is limited to the stomach and Ph. Currently, the Results section can be a bit repetitive, as the same analysis is repeated for each gut segment or question. Moreover, Ph changes beyond the stomach are not to be expected. Thus authors could consider prioritizing the Ph stomach results and moving other material to the supplements. Additionally, figure 4 which provides the most important results, could be improved, for instance, by present separately the scatter plot of disgust ratings vs stomach ph only for disgusting movies, as it is currently quite cluttered.

Since this is a new phenomena authors discovered with a relatively new methodology for the field, I believe authors should provide more evidence to validate and further characterize these findings. Right now the main results is backed up by a statistical model which shows a correlation across participants but tells little about the actual mechanisms underlying the phenomena. How long it takes acid to be secreted after a film? does it happen in each participant or only on some, how does it relate to tachygastria? These are some questions authors could tackle with their data to further convince their readers of the strength of their findings. Moreover, authors should also provide more info on the quality of the EGG recordings, and communicate to the reader that the lack of effects on the EGG could be due to the use of only one recording point in the stomach.

While the authors acknowledge that the evidence provided is correlational only, I think the manuscript would greatly benefit if a discussion of potential brain mechanisms and pathways was included explaining the mechanism linking the experience of disgust to secretion of acid in the stomach.

Finally, I understand that the results shed no insights on the discrete vs emotion debate, and this should be taken out from the discussion or further argumented.

*Reviewer #2 (Recommendations for the authors):*

– A little more detail about the measurements/analysis in the main text would be appreciated, if there is space – for example, has pH been normalised in Figure 4? Were any participants' datasets excluded from any of the measures (and if so new Ns should be reported)?

– I think there is probably one further analysis for the authors to run if they want to claim "these results were specific to the stomach" (page 15). That is a higher-level ANOVA that includes measurements from all 3 locations, and shows an interaction with location, justifying the separate analysis. I also think this would help a minor issue I had with respect to multiple comparison correction (below).

– The authors have used Bonferroni correction for multiple comparisons, which is great, but it was not clear to me exactly how many comparisons they were correcting for. As it stands, for the ph/pressure/temperature analyses, I think it should be at least 9 (3 locations x 3 measures per location), unless (as above) an initial ANOVA is run with all 3 locations, showing an interaction with location, in which case I think 3 is appropriate. This could be clarified on page 22-23 ('Statistical data analysis').

– It would be helpful if the authors included effect sizes of their main effects.

– Is there a (statistical) outlier in Figure 4. (top left – Disgusting) – the 'Happy' datapoint that has a much higher stomach pH than others (around 2.5 on the x axis)?

– I wonder if the authors might consider including one final post-hoc analysis simply examining valence (collapsing across negative vs positive emotions) and pH of the stomach. I think this would lend itself well to the Discussion section on specificity (or not) of gastric contribution to emotional state. This might be distinct from a concept of 'arousal' (all emotions).

– Was there a relationship between gastric sensations and pH/pressure/temperature in the same location of the GI tract? This might be useful to highlight with respect to its utility for future measures of gastric interoception.

*Reviewer #3 (Recommendations for the authors):*

-The Introduction would be more easily readable if it would have a paragraph structure.

-Several of the reported results are not central to the project and could be summarized (with full results in Supplementary materials). For example, the first two pages extensively report that the stimuli were perceived as intended (as also shown in previous studies using these stimuli). The interest here may be on the differences across sessions. If so, Session could be included as a factor in the ANOVA. (still, this would be a rather minor point compared to the subsequent analyses).

-The results are now mostly presented as a series of statistical tests, starting with "The Friedman ANOVA…". It would be helpful if the tests are motivated and explained, for example based on the hypotheses in the Introduction (e.g., "To test the hypothesis that participants would report higher gastric sensations.…, we compared subjective ratings of gastric sensations between.…"). Reporting the results in a more hypothesis-driven manner may also reduce the number of comparisons.

-Figure 4, the key results figure, could be improved to make a more convincing case for a relation between pH and Vas ratings (if it exists). The red dashed lines (Disgust) include dots, which cannot be distinguished from the actual data. The legend does not indicate what the dots represent (participants?) or what the regression lines and shading around it reflect. The axes labels are very small (particularly the lower three panels).

-The statistical analysis was not entirely clear to me; further motivation for the statistical approach would be welcome. The Results section refers to Model 1, but numbered models are not explicitly mentioned in the Methods, as far as I could tell. The other models (2 and 3) are not mentioned in the Results. Did you confirm your findings with other kinds of statistical tests, e.g., was there a simple correlation between gastric pH and disgust ratings?

-How did you deal with outliers (e.g., pH values in Figure 4)?

-Because the methods section is at the end, the results should be understandable without having read the methods section. This means that the measures and statistical models need to be explained in the Introduction or Results. For example, p.11 refers to "Model 1", but it is not clear what this model is. Similarly, it would be helpful if the procedure for the measures of "perceived visceral experience" (p.8) would be briefly explained.

-It would be relevant to know how objective and subjective measures of gastric changes relate to each other. Is gastric pH correlated with perceived gastric sensations? and, separately, is perceived gastric sensations correlated with subjective perception of emotions?

-Results are repeated on p.11: line 293-300 appear again on line 305-313. I think these should actually refer to different results (across items vs specific to disgust videos), correct?

---

## [Author Response]

Essential revisions (for the authors):Reviewer #1 (Recommendations for the authors):Below suggestions that I believe could improve the impact of the paper.1. The authors explore the link between multiple emotions, subjective reports, gut compartments, and multiple gut indices, but the main finding is limited to the stomach and Ph. Currently, the Results section can be a bit repetitive, as the same analysis is repeated for each gut segment or question. Moreover, Ph changes beyond the stomach are not to be expected. Thus authors could consider prioritizing the Ph stomach results and moving other material to the supplements.

As suggested by this Reviewer, we have relocated all the results concerning the small and large intestine to the supplementary materials (now called Appendix, following *eLife* rules). We hope this adjustment will reduce repetition and offer a more concise presentation of the main findings.

2. Additionally, figure 4 which provides the most important results, could be improved, for instance, by present separately the scatter plot of disgust ratings vs stomach ph only for disgusting movies, as it is currently quite cluttered.

Following this Reviewer’s and Editor’s suggestion we revised Figure 4 (now 5) and we hope that is now less cluttered and clearer. We put major emphasis on the main result related to the disgusting video clips, and we also revised the caption accordingly. We copied and pasted the revised figure below and inserted it in the revised main text.

3. Since this is a new phenomena authors discovered with a relatively new methodology for the field, I believe authors should provide more evidence to validate and further characterize these findings. Right now the main results is backed up by a statistical model which shows a correlation across participants but tells little about the actual mechanisms underlying the phenomena. How long it takes acid to be secreted after a film? does it happen in each participant or only on some, how does it relate to tachygastria? These are some questions authors could tackle with their data to further convince their readers of the strength of their findings.

We are aware that our results at the moment need further explorations and replications. To deal with this important remark, we:

Added a new analysis to check how our emotional inductions directly modulated the pH of the stomach (regardless of participants’ emotional ratings);In performing the analysis described in point 1, to also reduce the influence or more extreme values, we took into account all the available pH datapoints and included the effect of Time in the analysis.

In order to deal with the multiple data points collected from the same participant we conducted a linear mixed model analysis.

We added a paragraph with the results of this analysis in the Results section which on pages 15 and 16, line 411-446 now reads as follows:

“To explore how the emotional inductions could modulate the pH of the stomach and how the length of the exposure to that specific emotional induction could also play a role in modulating pH variations, we ran an additional model, Model 2. This model included all the pH datapoints registered using the Smartpill as dependent variable, the type of video clip and the number of the datapoints (“Time”) as fixed effects, and the by-subject intercepts as random effects (see Appendix 1 for a detailed description of the model). Model 2 had a marginal R^2^ = 0.014 and a conditional R^2^ = 0.79. Visual inspection of the plots did reveal some small deviations from homoscedasticity, visual inspection of the residuals did not show important deviations from normality. As for collinearity (tested by means of vif function of car package), all independent variables had a (GVIF^(1/(2*Df)))^2 < 10.

Type III analysis of variance of Model 2 showed a statistically significant main effect of the Time (F = 20.237, p < 0.001, Eta2 < 0.01) suggesting that independently from the type of video clip observed, the stomach pH significantly decreased as a function of the time of exposure to the induction.

A significant main effect of the type of video clip was also found (F = 22.242, p < 0.001, Eta2 = 0.01) suggesting that pH of the stomach changes when participants experienced different types of emotions. In particular, post hoc analysis revealed that pH was more acidic when participants observed disgusting compared to fearful (t = -11.417; p < 0.001), happy (t = -15.510; p < 0.001) and neutral (t = -3.598; p = 0.003) video clips.

Also, pH was more acidic when participants observed fearful compared to happy (t = -4.064; p < 0.001), and less acidic compared to neutral (t = 7.835; p < 0.001) and sad scenarios (t = 9.743; p < 0.001). Finally, pH was less acidic when participants observed happy compared to neutral (t = 11.923; p < 0.001). and sad videoclips (t = 13.806; p < 0.001), see Figure 6, left panel.

Interestingly, also the double interaction Time X Type of video clip was significant (F = 3.250, p = 0.0113, Eta2 < 0.01) suggesting that the time of the exposure to the induction differentially influenced the pH of the stomach depending on to the type of the observed video clip. Simple slope analysis showed that while pH did not change over time when observing disgusting (t = -1.2691; p = 0.2045) and happy (t = 0.4466; p = 0.6552) clips, it did significantly decrease over time when observing fearful (t = -4.4212; p < 0.001), sad (t = -2.0487; p = 0.0405) and neutral video clips (t = -2.7956; p = 0.0052), see Figure 6, right panel.”

We also changed the discussion accordingly (Page 20, lines: 557-564):

“To complement results showing a key role of the stomach pH in the perceived emotional experience, we also tested the hypothesis that independently from the conscious emotional experience (i.e. emotional VAS ratings) the pH varies during the observation of the different types of video clip. In fact, we found that stomach pH was generally more acidic when participants observed disgusting video clips and generally less acidic when participants observed happy video-clips. These effects did not change over time. Differently, we found that stomach pH was more acidic when participants observed fearful compared to happy video-clips, and less acidic compared to neutral and sad scenarios with a general decrease over time of the pH, see Figure 6.”

3) We included new evidence coming from different physiological markers that we did not analyse in the original manuscript. In fact the very same electrodes used to record the EGG signals also pick up the heart signal and thus we had the possibility to extract and analyse two markers of cardiac and autonomic nervous system activity, namely the heart rate (HR) and its variability (HRV). Specifically, we analysed if and how the root mean square successive difference between heartbeats (RMSSD) changed across the different emotional scenarios. According to the Task Force guidelines (1996), the RMSSD reflects the integrity of vagus nerve-mediated autonomic control of the heart. Therefore this analysis helped in discussing the effect of our emotional induction procedure in the non-gastric domain. In addition to that, we investigated the number of spontaneous eye blinks comparing them between the different emotions. In fact, besides hydrating the eyes and protecting them against foreign objects (e. g. Evinger, 1991, 1995) spontaneous blinking is a physiological mechanism that regulates information loss. It is linked to cognitive functions, particularly those related to attention, vigilance, and emotional states (e.g. Maffei & Angrilli, 2019; McIntire et al. 2014, Martin et al., 2015), and thus might be conceived as a measure of the salience triggered by the stimuli, in our case emotional video clips. To integrate results related to the HR, HRV and eye blinks we revised the main text as follows, Pages 12 and 13, Lines: 312-350:

“Autonomic and behavioural (oculomotor) markers of perceived emotions: heart rate (HR), heart rate variability (HRV), and spontaneous eye blinks

During the first session, i.e. when the capsule was in the stomach, we tested changes in heart rate (HR), heart rate variability (HRV), and number of spontaneous eye blinks in response to emotional experiences triggered by five types of video clip (disgust, fear, happiness, sadness, compared to neutral). To do this, after verifying via Kolmogorov-Smirnov tests that all data distributions were normally distributed (all ps> 0.05), we conducted three separate repeated-measures ANOVAs with type of video clip as a within-subjects factor and HR, HRV and number of eye blinks as dependent variables. We found a main effect of the type of video clip on HR (F(4, 112) = 5.652, p = 0.00035, η2 = 0.17) suggesting that participants’ HR changed across different emotions. Specifically, Bonferroni-corrected post-hoc analysis showed that HR was higher when participants observed the control neutral scenarios compared to the disgusting and fearful ones (all ps ≤ 0.008). All the other differences were not significant (all ps ≥ 0.124), see Figure 4, panel A for a graphical representation of the results.

Additionally, we found a significant main effect of the type of video clip on the RMSSD, (F(4, 112) = 2.658, p = 0.036, η2 = 0.09), suggesting changes in participants’ HRV across different emotions. Bonferroni-corrected post-hoc analysis revealed that HRV was higher when participants observed the disgusting video clips compared to the control neutral ones (p = 0.049), see Figure 4, panel B for a graphical representation of the results.

Regarding the spontaneous blinks, we found a main effect of the type of video clip (F(4, 108) = 12.371, p ≤ 0.0001, η2 = 0.32), suggesting that participants’ spontaneous eye blinks change across the different emotions. Bonferroni-corrected post-hoc analysis showed that participants blinked less when they observed the disgusting, fearful and happy video clips compared to the control neutral ones (all ps ≤ 0.009). Furthermore, they blinked more when they observed the sad videos compared to when they observed the disgusting and the fearful ones (all ps ≤ 0.001), but not when they observed the happy and neutral ones (all ps ≥ 0.228), see Figure 4, panel C for a graphical representation of the results.”

Finally, we also modified the introduction and the discussion to include these new results.

1. Electrophysiology, Task Force of the European Society of Cardiology and the North American Society of Pacing and Electrophysiology, T. F. O. T. E. S. O. C. T. N. A. S. O. P. (1996). Heart rate variability: standards of measurement, physiological interpretation, and clinical use. *Circulation*, *93*(5), 1043-1065

4. Moreover, authors should also provide more info on the quality of the EGG recordings, and communicate to the reader that the lack of effects on the EGG could be due to the use of only one recording point in the stomach.

To address this point, we (i) expanded our analysis to the tachygastric range and (ii) we ran a control analysis in an independent sample of participants (N = 25) of an ongoing study in which we used the same emotional induction procedure and we recorded the gastric rhythm from a 4-channels EGG electrodes.

Even with multiple channels we did not find convincing evidence that emotions differentially modulate the EGG frequency.

5. While the authors acknowledge that the evidence provided is correlational only, I think the manuscript would greatly benefit if a discussion of potential brain mechanisms and pathways was included explaining the mechanism linking the experience of disgust to secretion of acid in the stomach.

We thank the Reviewer for this important comment. Unfortunately, at the moment we do not have any data that may help to disclose the brain mechanisms underlying these effects. However, we are currently working on this important open issue. We added to the revised discussion a few speculation lines which are also copied and pasted below (Page 22, Lines: 635-645).

“Delving into the neural correlates underlying these phenomena, specifically focusing on the bidirectional pathways linking the stomach and the brain would also be of significant interest. For instance, investigating the potential role of brain regions like the anterior cingulate cortex (ACC) and the insula (AI), conceived as primary interoceptive cortices with functions encompassing homeostatic, emotional, limbic, and sensorimotor processes, would be valuable. Moreover, considering regions such as the nucleus tractus solitarii (NTS) in the brainstem, which serves as the pathway for major inputs projecting information to higher-order brain areas, could provide crucial insights (see the recent systematic review on the neural networks that underpin nausea in humans by Varangot-Reille et al., 2023). Additionally, it will be pivotal to map differences in stomach-brain coupling activity (Richter et al., 2017; Rebollo et al.,2018, 2022) during different emotional experiences.”

6. Finally, I understand that the results shed no insights on the discrete vs emotion debate, and this should be taken out from the discussion or further argumented.

As previously mentioned, we revised the discussion in order to better describe our results and toned down the interpretation that the present findings as directly supporting the discrete nature of emotions.

Please see the specific paragraph at page 21 lines 622-624 reads as follows and the revised discussion of the main text:

“Overall, and in line with theoretical and empirical evidence (Stephens et al., 2010, Damasio, 1999; Harrison et al., 2010; James, 1994, Lettieri et al., 2019; Stephens et al., 2010), our findings suggest that specific patterns of subjective, behavioural, and physiological measures are linked to unique emotional states.”

Reviewer #2 (Recommendations for the authors):1. A little more detail about the measurements/analysis in the main text would be appreciated, if there is space – for example, has pH been normalised in Figure 4?

We apologize for not reporting this information in the previous version of the manuscript. Yes, pH, temperature and pressure were mean-centred (and scaled) to be used as regressors in our models.

To make this point clear, we changed the Methods section (Page 30, Lines 861-868) that now reads as follows:

“To do so, three separate mixed-effects models (Model 1; Model 5 and Model 6) were run, one for each GI region (stomach, small bowel, and large bowel) having as dependent variable the VAS ratings relative to the emotional experience questionnaire, and as fixed factors emotion (induced by the different type of the video clip: disgust, fear, happy, neutral and sad) and item (the type of question participants were required to answer: perceived disgust, fear, happiness and sadness) as fixed factors, pH, temperature and pressure (mean-centred and scaled) as continuous fixed factors, and by-subject intercepts as random effects.”

We also modified the caption of Figure 5, that now reads as follows:

“Figure 5 Association between the Stomach pH and perceived emotions (3-way interaction between item, stomach pH, and type of video-clip). Association between stomach pH and perceived emotions (disgust, fear, happiness, and sadness) across the five categories of video clips (disgusting, fearful, happy, neutral, and sad). The figure highlights the main differences, primarily observed in the left section, concerning the disgusting video clips. Dots represent participants’ VAS ratings given during the different types of video clip, shadows indicate Confidence Intervals. The pH has been mean-centred and scaled. *p≤ 0.05; ** p≤ 0.01”

2. Were any participants' datasets excluded from any of the measures (and if so new Ns should be reported)?

Due to technical reasons ingestible data from one participant had to be excluded. Thus the analyses were performed in 30 out of 31 participants This information was already included in the Methods section, but we additionally clarified it. Page 28, lines 810 reads as follows:

“PH calibrations were performed correctly in all but one participant, therefore pH data from one participant were not considered in the statistical analyses since they were not reliable. Thus pH data were available for 30 out of 31 participants.”

For what concerns the EGG analysis, approximately 10% of the data was lost due to technical reasons or signal noise in certain experimental conditions/participants. This information has been included in the main text. Page 17, line: 457.

“Around the 10% of the were missed due to technical reasons or noise in the signal.”

3. I think there is probably one further analysis for the authors to run if they want to claim "these results were specific to the stomach" (page 15). That is a higher-level ANOVA that includes measurements from all 3 locations, and shows an interaction with location, justifying the separate analysis. I also think this would help a minor issue I had with respect to multiple comparison correction (below).

We thank the Reviewer for this comment. We opted to conduct separate analyses based on the district since the stomach and intestine exhibit macroscopic physiological differences owing to their distinct functions. For instance, the stomach's digestive function results in significantly different pH acidity compared to that characterizing the intestine. However, to explore the effect of different GI parts, we conducted a mixed-model ANOVA with pH as the dependent variable and district as the independent variable. The analysis revealed a significant and notably strong main effect of the district. Below, we copied and pasted the output of the analysis and the plot, indicating the Bonferroni corrected significant differences. These differences were attributed to distinct pH values across the three districts. The most acidic values characterize the stomach. Moreover, we are aware of this intrinsic limitation, and in order to deal with the Reviewer’s comment, we have revised the main text and eliminated sentences that might imply a comparison between the three districts.

**Author response table 1. sa2table1:** Type III Analysis of Variance Table with Satterthwaite's method.

	Sum Sq	Sum Sq	NumDF	DenDF F value	Pr(>F)
district	1509.5	754.74	2 1681.4	9023	< 2.2e-16 ***

**Author response image 1. sa2fig1:** 

4. The authors have used Bonferroni correction for multiple comparisons, which is great, but it was not clear to me exactly how many comparisons they were correcting for. As it stands, for the ph/pressure/temperature analyses, I think it should be at least 9 (3 locations x 3 measures per location), unless (as above) an initial ANOVA is run with all 3 locations, showing an interaction with location, in which case I think 3 is appropriate. This could be clarified on page 22-23 ('Statistical data analysis').

As we wrote in the manuscript we ran three separate mixed models ANOVA one per district that, as we motivated above, we did not want directly compare due to the differences of the three environments inherently linked to their different functions. By doing that, we found that only in the stomach the perceived emotions were predicted by pH changes (we replicated this analysis with a similar and more robust approach showing how ph changes in relation to the different emotional inductions). In order to test which slope was significantly different from zero we used the sim_slopes function of the interaction package (Long, 2019). Then, to check and avoid false positives we performed, by using emtrends function of the emmeans R package (Lenth et al., 2019), post hoc tests using the more conservative model (the one with the condition as random slope); and automatically corrected (namely for the different condition and perceived emotions) for Bonferroni by using the “adjst” argument and by specifying Bonferroni as correction. We reported the detailed statistics and results in the main text and in the supplementary materials (now called Appendix, following *eLife* rules).

5. It would be helpful if the authors included effect sizes of their main effects.

We thank this Reviewer for this comment. We added the effect size of our main and interaction effects. They can be found track-changed in the Main Text as well as in the Supplementary Information (now called Appendix, following eLife rules).

6. Is there a (statistical) outlier in Figure 4. (top left – Disgusting) – the 'Happy' datapoint that has a much higher stomach pH than others (around 2.5 on the x axis)?

We acknowledge that our sample is not that big, and that data present variability. Therefore we followed a series of steps to be more conservative.

Firstly, as we already indicated in the original version of the manuscript we provided additional evidence supporting our main results by using mixed function of afex R package (Singmann et al., 2015) and calculated p-values by using a more robust method namely, bootstrap (Efron, 1979, Ann Statist 7:1–26) with 1000 reiterations. Results confirm our previous findings. We used Bonferroni correction in case of significant interactions.

Furthermore, we confirmed our main results, those relating stomach pH with the perception of disgust, fearful and happy induced by disgusting video clips, with a robust linear mixed model by also including the pH slope as random effect. To do that we used rlmer function of the Robustlmm package (Koller, M. (2014). Robustlmm: Robust estimating equations and examples. Cran R-project. org.). Due to the fact that it is not possible to run an ANOVA on the robust model and to the singularity of the fit we did not include this analysis in the main text.

Finally, in the revised version of the manuscript we added novel evidence to our main results and a new analysis. In particular, we used all the datapoints recorded from the ingestible device and we performed a mixed models analysis with pH as dependent variable, type of video clip and number of datapoints as fixed factors, and the by-subject intercepts as random effects. We replicated the main analysis, this time without the influence of more extreme values.

Results of this analysis are described in point 1.7, below and, can be found in track change in the manuscript at Page 15&16, lines: 411-446 in the main text.

“To explore how the emotional induction could modulate the pH of the stomach and how the length of the exposure to that specific emotional induction could also play a role in modulating pH variations, we ran an additional model, Model 2. This model included all the pH datapoints registered using the Smartpill as dependent variable, the type of video clip and the number of the datapoints (“Time”) as fixed effects, and the by-subject intercepts as random effects (see Appendix 1 for a detailed description of the model). Model 2 had a marginal R^2^ = 0.014 and a conditional R^2^ = 0.79. Visual inspection of the plots did reveal some small deviations from homoscedasticity, visual inspection of the residuals did not show important deviations from normality. As for collinearity (tested by means of vif function of car package), all independent variables had a (GVIF^(1/(2*Df)))^2 < 10.

Type III analysis of variance of Model 2 showed a statistically significant main effect of the Time (F = 20.237, p < 0.001, Eta2 < 0.01) suggesting that independently from the type of video clip observed, the stomach pH significantly decreased as a function of the time of exposure to the induction.

A significant main effect of the type of video clip was also found (F = 22.242, p < 0.001, Eta2 = 0.01) suggesting that pH of the stomach changes when participants experienced different types of emotions. In particular, post hoc analysis revealed that pH was more acidic when participants observed disgusting compared to fearful (t = -11.417; p < 0.001), happy (t = -15.510; p < 0.001) and neutral (t = -3.598; p = 0.003) video clips.

Also, pH was more acidic when participants observed fearful compared to happy (t = -4.064; p < 0.001), and less acidic compared to neutral (t = 7.835; p < 0.001) and sad scenarios (t = 9.743; p < 0.001). Finally, pH was less acidic when participants observed happy compared to neutral (t = 11.923; p < 0.001). and sad videoclips (t = 13.806; p < 0.001), see Figure 6, left panel.”

Interestingly, also the double interaction Time X Type of video clip was significant (F = 3.250, p = 0.0113, Eta2 < 0.01) suggesting that the time of the exposure to the induction differentially influenced the pH of the stomach depending on to the type of the observed video clip. Simple slope analysis showed that while pH did not change over time when observing disgusting (t = -1.2691; p = 0.2045) and happy (t = 0.4466; p = 0.6552) clips, it did significantly decrease over time when observing fearful (t = -4.4212; p < 0.001), sad (t = -2.0487; p = 0.0405) and neutral video clips (t = -2.7956; p = 0.0052), see Figure 6, right panel.

7. I wonder if the authors might consider including one final post-hoc analysis simply examining valence (collapsing across negative vs positive emotions) and pH of the stomach. I think this would lend itself well to the Discussion section on specificity (or not) of gastric contribution to emotional state. This might be distinct from a concept of 'arousal' (all emotions).

Thanks for this insightful and cogent comment which we are happy to take into consideration. However, we need to firstly point out that we have three negative emotions (fear, disgust and sad) and only one positive emotion (happy), therefore the observations will be not balanced. Additionally, we did not inquire about perceived valence but instead utilized discrete emotional ratings. At any rate, to comply with this potentially important request, we categorized the experimental conditions into three groups: negative valence video clips (fear, disgust, and sadness), positive valence video clips (happiness), and neutral ones (control). The analysis did not reveal any significant double or triple interactions with the pH. This suggests that pH may play a specific role in distinct emotional experiences and may not be inherently linked to negative valence alone.

Given the already mentioned methodological limitations of this analysis, if the referee agrees, we would prefer not to include it in the manuscript.

**Author response table 2. sa2table2:** anova(Stomach_VASvalence,type=3,ddf="Satterthwaite") Type III Analysis of Variance Table with Satterthwaite's method.

	Sum Sq	Mean Sq	NumDF	DenDF	F value	Pr(>F)
valence	12073	12072.7	1	507.44	23.6486	1.545e-06 ***
item	43487	14495.8	3	502.15	28.3949	< 2.2e-16 ***
zpressure	832	832.5	1	414.53	1.6307	0.20232
zpH	1585	1585.0	1	191.59	3.1047	0.07911
zT	1277	1276.6	1	107.39	2.5007	0.11673
valence:item	69907	23302.4	3	502.15	45.6456	< 2.2e-16***
valence: zpressure	253	252.6	1	520.36	0.4947	0.48213
valence:zpH	762	761.9	1	508.95	1.4925	0.22240
valence:zT	691	690.9	1	506.45	1.3534	0.24523
item:zpressure	1810	603.5	3	502.15	1.1821	0.31596
item:zpH	1736	578.7	3	502.15	1.1336	0.33493
item:zT	375	125.0	3	502.15	0.2448	0.86503
valence: zpressure	2025	675.1	3	502.15	1.3225	0.26627
valence:zpH	130	43.5	3	502.15	0.0852	0.96814
valence:zT	200	66.7	3	502.15	0.1307	0.94183

8. Was there a relationship between gastric sensations and pH/pressure/temperature in the same location of the GI tract? This might be useful to highlight with respect to its utility for future measures of gastric interoception.

We thank this Reviewer for the highly valuable comment. To deal with it, we ran an additional exploratory analysis to check whether subjective gastric sensations were associated with changes of gastric pH, temperature and pressure. To test for the presence of this association, we built a model similar to the one used for testing the relationship between pill data (i.e. gastric pH, temperature and pressure) and perceived emotions (i.e. VAS ratings). This model had as dependent measure the 0-100 VAS ratings and the Type of item (type of perceived visceral sensations: cardiac, respiratory and gastric) as fixed effects, and pH/temperature and pressure of the stomach as covariates. We only found a significant and potentially interesting double interaction between Type of item X stomach pH (F(2, 333.26) = 4.6049, p = 0.01065). However, when we ran post-hoc tests, namely simple slopes analyses, no effect turned out to be statistically significant.

Reviewer #3 (Recommendations for the authors):1. The Introduction would be more easily readable if it would have a paragraph structure.

As suggested by this Reviewer, we added a paragraph structure to the introduction which is now divided in three paragraphs, namely:

1) The neglected role of gastrointestinal (GI) system in emotions

2) Ingestibles as an innovative approach for exploring the deep physiology of emotions

3) Hypotheses underlying this study

We hope that this change made more easily readable the Introduction of our study.

2. Several of the reported results are not central to the project and could be summarized (with full results in Supplementary materials). For example, the first two pages extensively report that the stimuli were perceived as intended (as also shown in previous studies using these stimuli). The interest here may be on the differences across sessions. If so, Session could be included as a factor in the ANOVA. (still, this would be a rather minor point compared to the subsequent analyses).

We understand that the way in which stimuli are perceived is not the central point; however, we think is important to provide data on the perceived emotions related to the different categories of video clips and most importantly on the visceral sensations that they triggered. We agree that the Results section might be improved because it presented many repetitions who distracted the readers. To take into account the points made by this Reviewer (and by Reviewer#1), we decided to move all the analysis concerning small and large bowel in the supplementary materials (now called Appendix, following *eLife* rules) in order to avoid redundancy and improve the clarity.

3. The results are now mostly presented as a series of statistical tests, starting with "The Friedman ANOVA…". It would be helpful if the tests are motivated and explained, for example based on the hypotheses in the Introduction (e.g., "To test the hypothesis that participants would report higher gastric sensations.…, we compared subjective ratings of gastric sensations between.…"). Reporting the results in a more hypothesis-driven manner may also reduce the number of comparisons.

We thank this Reviewer for this comment. To deal with it, we now presented the results with an initial sentence that motivates the statistical test and make explicit the hypothesis.

Page 6 Lines 171-175

“To verify the effectiveness of our video clips in eliciting distinct emotional states, we conducted a Friedman ANOVA to compare the perceived emotional experiences (i.e. disgust, fear, happiness and sadness) triggered by the five types of video clip (varying for their content, i.e. disgusting, fearful, happy, sad, and neutral) during the first session (namely when the capsule was in the stomach). In agreement with the hypotheses, we found that Friedman ANOVA etc.”

Page, 9 lines 233-239

“To test the hypothesis that the perceived visceral experience (i.e. gastric, cardiac, respiratory sensations and arousal) varied according to the emotional state triggered by the five types of video clip, and specifically that gastric sensations characterize more disgust and happiness, as suggested by literature (Nummenmaa et al., 2014), we performed a Friedman ANOVA. The Friedman ANOVA was statistically significant (χ^2^(19) = 323.399; p < 0.0001), suggesting that participants perceived different visceral sensations after observing the different content of the video clips.”

Page, 12 lines 314-320

**“**During the first session, i.e. when the capsule was in the stomach, we tested changes in heart rate (HR), heart rate variability (HRV), and number of spontaneous eye blinks in response to emotional experiences triggered by five types of video clip (disgust, fear, happiness, sadness, compared to neutral). To do this, after verifying via Kolmogorov-Smirnov tests that all data distributions were normally distributed (all ps> 0.05), we conducted three separate repeated-measures ANOVAs with type of video clip as a within-subjects factor and HR, HRV and number of eye blinks as dependent variables. “

Page, 13 lines 354-358

“To test the main hypothesis underlying this study, namely that the emotional experience triggered by the five categories of the video-clips is linked to the inner physiology of the stomach, we conducted a mixed model analysis (Model 1, see Data analysis session and Appendix 1 for details) with emotional VAS score as dependent variable, type of video clip and item as independent factors, and stomach pH, temperature and pressure as covariates.”

Page, 15 lines 411-416

“To explore how the emotional induction could modulate the pH of the stomach and how the length of the exposure to that specific emotional induction could also play a role in modulating pH variations, we ran an additional model, Model 2 (see Data analysis session and Appendix 1 for details). This model included all the pH datapoints registered using the Smartpill as dependent variable, the type of video clip and the number of the datapoints (“Time”) as fixed effects, and the by-subject intercepts as random effects (see Appendix 1 for a detailed description of the model).”

Page 17, lines 451-456

“With respect to the hypothesis that emotional experience might be linked also to the electromyographic activity of the stomach (EGG) we ran Model 3 and 4 (see above, Data analysis paragraph). These models contained the emotional VAS ratings as dependent variables, the type of video clip and items as independent factors, the individual normogastric and tachygastric EGG peak frequency respectively as covariates and the by-subject intercepts as random effects.”

Page 18, lines 486-490

“To test the more exploratory hypothesis that the emotional experience triggered by the five categories of the video clip is linked to the inner physiology of the small and large bowel, we conducted two separate mixed model analysis (Model 5 and Model 6) with VAS ratings as dependent variable, type of video clip and item as independent factors, and small and large bowel pH, temperature and pressure respectively as covariates.”

4. Figure 4, the key results figure, could be improved to make a more convincing case for a relation between pH and Vas ratings (if it exists). The red dashed lines (Disgust) include dots, which cannot be distinguished from the actual data. The legend does not indicate what the dots represent (participants?) or what the regression lines and shading around it reflect. The axes labels are very small (particularly the lower three panels).

We apologize if the figure was not clear enough, the other Reviewers raised very similar concerns about it. To improve Figure 4 (now 5) we modified the original plot (see the reply to a similar point raised by Reviewer 1), hoping that now results are clearer. In specific, we enlarged the main panel as suggested by Reviewer#1, increased the font of the x-y axes, and revised the legend, that now reads as follows:

*“*Figure 5 Association between the Stomach pH and perceived emotions (3-way interaction between item, stomach pH, and type of video-clip). This figure illustrates the association between stomach pH and perceived emotions (disgust, fear, happiness, and sadness) across the five categories of video clips (disgusting, fearful, happy, neutral, and sad). Notably it highlights the main differences, primarily observed in the left panel, concerning the disgusting video clips. Panel on the upper right highlights the main differences concerning the fearful video clips. Dots represent participants’ VAS ratings following the different types of video clip, shadows indicate Confidence Intervals. The pH has been mean-centred and scaled. *p≤ 0.05; ** p≤ 0.01”

5. The statistical analysis was not entirely clear to me; further motivation for the statistical approach would be welcome. The Results section refers to Model 1, but numbered models are not explicitly mentioned in the Methods, as far as I could tell. The other models (2 and 3) are not mentioned in the Results. Did you confirm your findings with other kinds of statistical tests, e.g., was there a simple correlation between gastric pH and disgust ratings?

In order to clarify the analysis, as suggested in you previous 3.4 comment we added lines explaining the aim of the analyses. Furthermore, we also checked that all Models were clearly described in the main text or in the Supplementary materials (now called Appendix, following *eLife* rules) and with the models the corresponding results. Importantly, to deal with the comments raised by this Reviewer and to take into account similar points raised by Reviewer#1 and #2, we performed additional analysis to support our claims and hope that the added evidence has made this work more robust and reliable. Please see our reply to the first point (3.1) containing the description of the additional analyses, the corresponding results and the plot of the novel results.

6. How did you deal with outliers (e.g., pH values in Figure 4)?

A similar point was raised also by the other Reviewers and as we stated above, we are aware that our results at the moment need further explorations and replications, but we also think that they should be described to highlight the impact of the gastric signals on higher-order functions by taking into account all the recent evidence on the coupling between the stomach and brain and the impact of gastric signals in non-food/nutrition-related functions (e.g. studies by Rebollo, Catherin Tallon Baudry et al.; Khalsa et al.; Avidan et al., Critchley et al. etc.).

To specifically reduce the influence of more extreme values, we:

Added a new analysis to check how our emotional induction directly modulated the pH of the stomach, regardless of participants’ emotional ratings. This approach significantly reduced the influence of extreme values and supported the original findings because it included not only the mean value but all the collected datapoints. Consistently with the previous finding, the new results support the modulation of the pH due to the emotional induction, indicating that the observation of disgusting video clips causes more acidic pH, while observation of happy video clips causes less acidic pH. To better visualize the effects we added another plot (Figure 5);Mitigated the effect of false positives and attenuating the impact of extreme values through the application of a function from the afex R package (Singmann et al., 2015) which allows the computation of p-values through the robust bootstrap method (with 1000 reiterations) for mixed models (Efron, 1979, Ann Statist 7:1–26). Furthermore, post hoc tests were conducted using the emtrends function of the emmeans R package (Lenth et al., 2019) on the more conservative model (with “Type of video” as a random slope). Moreover, Bonferroni correction was applied for all post hoc comparisons.Confirmed our main results, those relating stomach pH with the perception of disgust, fearful and happy induced by disgusting video clips, with a robust linear mixed model by also including the pH slope as random effect. To do that we used rlmer function of the Robustlmm package (Koller, M. (2014). Robustlmm: Robust estimating equations and examples. Cran R-project. org.). Due to the fact that it is not possible to run an ANOVA on the robust model and to the singularity of the fit we did not include this analysis in the main text.Adopted a parsimonious stance, we also acknowledged in the paper that further studies are necessary to corroborate these findings conclusively.

The new paragraph with the results of the novel analysis is described in the Results section which on Page 15&16, lines: 411-446 now reads as follows:

“To explore how the emotional induction could modulate the pH of the stomach and how the length of the exposure to that specific emotional induction could also play a role in modulating pH variations, we ran an additional model, Model 2. This model included all the pH datapoints registered using the Smartpill as dependent variable, the type of video clip and the number of the datapoints (“Time”) as fixed effects, and the by-subject intercepts as random effects (see Appendix 1 for a detailed description of the model). Model 2 had a marginal *R^2^* = 0.014 and a conditional *R^2^* = 0.79. Visual inspection of the plots did reveal some small deviations from homoscedasticity, visual inspection of the residuals did not show important deviations from normality. As for collinearity (tested by means of *vif* function of *car* package), all independent variables had a (GVIF^(1/(2*Df)))^2 < 10.

Type III analysis of variance of Model 2 showed a statistically significant main effect of the Time (F = 20.237, *p* < 0.001, Eta2 < 0.01) suggesting that independently from the type of video clip observed, the stomach pH significantly decreased as a function of the time of exposure to the induction.

A significant main effect of the type of video clip was also found (F = 22.242, *p* < 0.001, Eta2 = 0.01) suggesting that pH of the stomach changes when participants experienced different types of emotions. In particular, post hoc analysis revealed that pH was more acidic when participants observed disgusting compared to fearful (t = -11.417; *p* < 0.001), happy (t = -15.510; *p* < 0.001) and neutral (t = -3.598; *p* = 0.003) video clips.

Also, pH was more acidic when participants observed fearful compared to happy (t = -4.064; *p* < 0.001), and less acidic compared to neutral (t = 7.835; *p* < 0.001) and sad scenarios (t = 9.743; *p* < 0.001). Finally, pH was less acidic when participants observed happy compared to neutral (t = 11.923; *p* < 0.001). and sad videoclips (t = 13.806; *p* < 0.001), see Figure 6, left panel.

Interestingly, also the double interaction Time X Type of video clip was significant (F = 3.250, *p* = 0.0113, Eta2 < 0.01) suggesting that the time of the exposure to the induction differentially influenced the pH of the stomach depending on to the type of the observed video clip. Simple slope analysis showed that while pH did not change over time when observing disgusting (t = -1.2691; *p* = 0.2045) and happy (t = 0.4466; *p* = 0.6552) clips, it did significantly decrease over time when observing fearful (t = -4.4212; *p* < 0.001), sad (t = -2.0487; *p* = 0.0405) and neutral video clips (t = -2.7956; *p* = 0.0052), see Figure 6, right panel.”

7. Because the methods section is at the end, the results should be understandable without having read the methods section. This means that the measures and statistical models need to be explained in the Introduction or Results. For example, p.11 refers to "Model 1", but it is not clear what this model is. Similarly, it would be helpful if the procedure for the measures of "perceived visceral experience" (p.8) would be briefly explained.

We know that having the Results before the Methods might be confounding, however this is requested by the formatting guidelines of *eLife* journal. As suggested by this Reviewer we now introduced a descriptive sentence before the results and we now hope that they are clearer.

8. It would be relevant to know how objective and subjective measures of gastric changes relate to each other. Is gastric pH correlated with perceived gastric sensations? and, separately, is perceived gastric sensations correlated with subjective perception of emotions?

We thank also Reviewer#3 for its constructive input which echoes a similar one raised by Reviewer#2. To deal with the first comment raised by this Reviewer, we ran an additional exploratory analysis to check whether subjective gastric sensations (i.e. subjective gastric measures) were associated with changes of gastric pH, temperature and pressure (i.e. objective gastric measures). To test for the presence of this association, we built a model similar to the one used for testing the relationship between pill data (i.e. gastric pH, temperature and pressure) and perceived emotions (i.e. VAS ratings). This model had as dependent measure the 0-100 VAS ratings and the Type of Item (type of perceived visceral sensations: cardiac, respiratory and gastric) as fixed effects, and pH/temperature and pressure of the stomach as covariates. We only found a significant and potentially interesting double interaction between Type of item X stomach pH (F(2, 333.26) = 4.6049, p = 0.01065). However, when we ran post-hoc tests, namely simple slopes analyses, no effect turned out to be statistically significant.

As to the last point i.e. running correlation analyses between self-reported gastric sensations and self-reported perceived emotions. To comply with the request we ran exploratory correlation analysis within the emotions that have been found to be linked to the gastric activity, i.e. disgust, fear and happy. We did not find a general association between the two variables, rather a specific a positive relationship between perceived disgust and perceived gastric sensation (r = 0,4945; p = 0. 0064) suggesting that the higher the gastric sensations the higher the perception of disgust. The same correlation was not significant for happy and fear clips (all r < (abs) 0.131; all ps > 0.499). If the Reviewer thinks that this analysis should be reported in the main or supplementary materials (now called Appendix, following *eLife* rules) we will be happy to include it.

9. Results are repeated on p.11: line 293-300 appear again on line 305-313. I think these should actually refer to different results (across items vs specific to disgust videos), correct?

Yes, correct they are different results. The first refer to the double interaction (pH * item), namely how VAS ratings changed with respect to the pH and the type of question participants were requested to answer, thus independently from the type of video participants observed. The second refer to the triple interaction (pH * item * emotion contained in the video clips).

To avoid ambiguity, we added two sentences in yellow below, and in track-change in the main text and we now hope that results are presented in a clear way.

Page 13 and 14, lines 364-386

“Type III analysis of variance of Model 1 showed a statistically significant 2-way interaction between item (i.e. perceived disgust, fear, happiness and sadness) and gastric pH (F = 8.214, p < 0.0001, bootstrap p-value < 0.001, Eta2 (partial) = 0.05, Appendix 1, Figure 5), suggesting that the emotional experience reported by participants on the VAS ratings, irrespective of the type of observed video clip, varied according to the pH of the stomach and the type of perceived emotion. Specifically, the follow-up post hoc simple slope analysis, conducted to explore the significant 2-way interaction (gastric pH*item), showed that the lower (i.e., more acidic) was the pH of the stomach, the more our participants reported to feel disgusted and afraid, while the higher the pH of their stomach (i.e., less acidic), the more they reported to feel happy.

The 2-way interaction was further defined by a 3-way interaction between video-clip content, item, and pH (F = 1.978, p = 0.025, bootstrap p-value = 0.026, Eta2 (partial) = 0.05, Figure 5), suggesting that emotional experience reported by participants on the VAS ratings varied also as a function of the content of the projected video clips, besides the pH of their stomach and the type of emotion perceived. Follow-up post hoc simple slopes analysis, performed to explain the significant 3-way interaction (gastric pH*item*type of video clip), showed that the lower (i.e. more acidic) was the pH of the stomach, the more our participants reported to feel disgusted and afraid, after observing disgusting video clips, while the higher was the pH of their stomach (i.e. less acidic) the more reported to feel happy.”